# G-AlignNet: Geometry-Driven Quality Alignment for Robust Dynamical Systems Modeling

## Abstract

The Neural ODE family has shown promise in modeling complex systems but often assumes consistent data quality, making them less effective in real-world applications with irregularly sampled, incomplete, or multi-resolution data. Current methods, such as latent ODEs, aim to address these issues but lack formal performance guarantees and can struggle with highly evolving dynamical systems. To tackle this, we propose a novel approach that leverages parameter manifolds to improve robustness in system dynamical modeling. Our method utilizes the orthogonal group as the underlying structure for the parameter manifold, facilitating both quality alignment and dynamical learning in a unified framework. Unlike previous methods, which primarily focus on empirical performance, our approach offers stronger theoretical guarantees of error convergence thanks to the well-posed optimization with orthogonality. Numerical experiments demonstrate significant improvements in interpolation and prediction tasks, particularly in scenarios involving high- and low-resolution data, irregular sampling intervals, etc. Our framework provides a step toward more reliable dynamics learning in changing environments where data quality cannot be assumed.

## 1 Introduction

Learning accurate dynamical models for decision-making, control, and Reinforcement Learning (RL) in complex systems has emerged as a key challenge, particularly in environments where data is of inconsistent quality and system dynamics are subject to continuous adaptation. Most existing approaches fail to maintain robust performance due to their reliance on high-quality data (Nagabandi et al., 2018). Real-world scenarios, such as power grids, healthcare, and transportation networks, often produce heterogeneous data with varying resolutions, incomplete observations, and inconsistent sampling rates, complicating the modeling task (Tuballa & Abundo, 2016; Zhu et al., 2018). For instance, in smart grids, Phasor Measurement Units (PMUs) provide high-resolution data, while lower-resolution sensors like Remote Terminal Units (RTUs) are used to reduce communication and infrastructure costs (Li et al., 2024b).

In control and engineering systems, the most severe and persistent issue is *data incompleteness*, referring to missing values in datasets, which can be categorized into: (1) Low-Resolution (LR) measurements, caused by LR sensors (Li et al., 2024a) or downsampling to meet communication constraints (Willett et al., 2011); (2) Periodic data losses due to communication or sensor failures, external events, etc. (Gill et al., 2011); (3) Random data losses (e.g., irregular sampling (Kidger et al., 2020; Chen et al., 2024)), arising from sensor configurations, data corruption, or human errors (Kundu & Quevedo, 2021). These scenarios are illustrated in Appendix A. Our study focuses on addressing *data incompleteness*. Other data quality issues, such as *data inaccuracy* and *inconsistency*, can often be addressed using established techniques (Chen & Abur, 2006), effectively removing bad data and reducing them to data incompleteness. Also, we restrict our analysis to systems with low nonlinearity and limited noise, prioritizing the challenge of handling significant missing data.

While data imputation techniques and sequence models, such as Recurrent Neural Networks (RNN), have been proposed, they often lack performance guarantees and struggle in highly dynamic environments (Kong et al., 2013). A better option is to leverage Ordinary Differential Equation (ODE) solvers for continuous-time evaluations, e.g., the family of Neural ODEs (Chen et al., 2018; Kidger et al., 2020; Rubanova et al., 2019). These methods still face limitations in providing robust guarantees when data quality is inconsistent. This gap calls the need for a more structured and theoretically grounded approach that can directly address the problem of mixed-quality data in evolving systems.

To this end, we propose **G-AlignNet**, a novel framework that leverages parameter geometry on the orthogonal group to enhance learning dynamics and provide robust performance guarantees. Unlike traditional data manifold approaches (Li & Zhao, 2021; Li et al., 2024b) that are sensitive to different data quality issues, G-AlignNet operates on a parameter manifold, ensuring adaptability and alignment between high- and low-quality data through geometric optimization. The orthogonal group structure not only enables continuous adaptation using analytical Lie algebra (Helgason, 1978) but also provides tight theoretical guarantees for error convergence, which is better than other manifold-based signal recovery (Chen et al., 2010). In particular, the built-in orthogonality in G-AlignNet brings a well-posed on-manifold optimization that leads to globally optimal solutions for aligning high- and low-quality data (Banica & Speicher, 2009; Choromanski et al., 2020a). This ensures stable learning even in the presence of many missing data or highly irregular sampling rates.

In summary, our contributions are as follows:

- **Introducing G-AlignNet**: A novel geometric framework that unifies the modeling of high- and low-quality data through parameter manifolds, offering robust adaptation and data imputation capabilities. G-AlignNet is applicable to many base models, such as RNNs, Implicit Neural Representations (INRs), and Physics-Informed Neural Networks (PINNs).

- **Performance Guarantees**: We establish theoretical convergence guarantees for interpolation tasks, demonstrating significant improvements over existing methods.

- **Empirical Validation**: We demonstrate that G-AlignNet outperforms state-of-the-art models across multiple domains, particularly in settings with mixed-resolution data, missing observations, and varying sampling rates.

G-AlignNet sets a new direction in the field of neural ODEs by introducing a principled, geometry-based approach that addresses the critical issue of data quality in complex systems, offering a more reliable foundation for decision-making and control in evolving environments.

## 2 RELATED WORK

**Neural ODEs and Irregular Data**. Neural ODE families have gained widespread attention for modeling continuous-time dynamics in deep learning frameworks (Chen et al., 2018). These models have been applied across various domains due to their flexibility in handling time series data. Extensions such as Latent ODEs (Rubanova et al., 2019), Neural Controlled Differential Equations (Neural CDEs) (Kidger et al., 2020), and stochastic Neural ODEs (Li et al., 2020) have addressed irregular sampling. However, this doesn't necessarily mean that all data incompleteness issues in the above categories (1) ∼ (3) can be fully addressed. Significant data losses, such as low-resolution data, inherently lead to insufficient dynamic information for learning. For example, as shown in our theoretical analysis in Section 3.3, learning ODE dynamics can be analyzed through the framework of perturbed IVPs (Hillebrecht & Unger, 2022) with the accumulation of truncation and round-off errors, which is large with significant data losses.

**Data Imputation to Pre-Process Low-Quality Data**. To address this information gap, data imputation techniques are employed to enhance data quality before using Neural ODE-based methods. These techniques leverage prior knowledge, explicit assumptions about the system's behavior, or relevant high-quality data streams to reconstruct the missing information and enhance the learning process. Model-based methods, such as multidimensional interpolation (Habermann & Kindermann, 2007) and physical model-based estimations (Sacchi et al., 1998), rely on explicit assumptions about system behavior. Optimization-based techniques, including Compressed Sensing (Donoho, 2006), matrix completion, and Bayesian methods (Yi et al., 2023), frame imputation as minimizing a loss function by assuming low-rank or sparsity structures. Signal processing and machine learning models offer data-driven solutions that can adapt to complex patterns (Fukami et al., 2021; Li et al., 2024a), yet these often overlook domain-specific structures. Despite their utility, many existing approaches are inconsistent with the underlying data structure, as they rely on simplifying assumptions that fail to capture the intrinsic dynamics of complex systems.

**Manifold Learning for Dynamical Systems**. Manifold learning has long been used to represent high-dimensional data on lower-dimensional structures, allowing models to learn the intrinsic geometry of the data (Tenenbaum et al., 2000; Roweis & Saul, 2000). Common methods include

discrete graph-based approximations (Wang et al., 2018b;a; Li & Zhao, 2021) and continuous flows (Cui et al., 2014; Li et al., 2024b). The latter is well-suited for dynamical modeling, especially for continuous systems. For example, several methods based on Neural ODE have been employed to capture the dynamical data flows (Asikis et al., 2022; Legaard et al., 2023; Koenig et al., 2024; Chi, 2024). However, Neural ODEs are unsuitable for adaptive systems with complex data manifolds.

Hence, recent methods model a relatively simple parameter manifold of a DL model and allow the parameters to adapt across different time intervals. Specifically, (Du et al., 2021) uses parameter graph for approximation, while (Chalvidal et al., 2020; Yin et al., 2022; Choromanski et al., 2020b; Cho et al., 2024) leverage another Neural ODE to generate on-manifold parameter flows. However, their flow generations, without any restrictions, are sensitive to the quality of data. (Choromanski et al., 2020b) is the most relevant work to ours and introduces an orthogonal group to improve training stability. We give a more generalized framework to process sequential measurements and link this representation to geometric optimizations, providing provable quality alignment guarantees. This model with well-structured geometry properties has significant potential for domains like on-manifold RL (Liu et al., 2022; 2024; Ammar et al., 2015).

**Geometric Optimization on Orthogonal Groups**. Geometric optimization has become a valuable tool in machine learning, particularly for ensuring stability and optimizing over structured spaces like the orthogonal group (Boumal, 2020; Choromanski et al., 2020a). This optimization, which is well-studied in Lie group theory (Helgason, 1978) and Riemannian geometry (Absil et al., 2009), allows for the preservation of critical geometric properties such as orthogonality and invariance, which lead to more robust learning. Applications in deep learning, including orthogonalization techniques for stable training (Huang et al., 2018), provide inspiration for our method, which leverages these properties to ensure globally optimal solutions for aligning high- and low-quality data. Our approach builds on these methods to offer a closed-form solution with guaranteed performance improvements.

## 3 METHODOLOGY

### 3.1 PROBLEM FORMULATION

We aim to devise a model that can learn from both high-quality (HQ) and low-quality (LQ) data, aligning their quality and making accurate predictions. Let $s(t_i)$ represent the state of the system at time $t_i$. A learning model $f_\Theta(s(t_i))$, parameterized by $\Theta$, predicts the future state $s(t_{i+1})$. The model can be generalized to a probabilistic model $\hat{p}_\Theta(\cdot)$ with a focus on the geometry of $\Theta$.

Let $\{\boldsymbol{x}(t_i)\}_{i \in \mathcal{N}_x}$ represent HQ measurements and $\{\boldsymbol{y}(t_i)\}_{i \in \mathcal{N}_y}$ represent LQ measurements, where $\boldsymbol{x} \in \mathbb{R}^{d_x}$, $\boldsymbol{y} \in \mathbb{R}^{d_y}$, and $\boldsymbol{s} = [\boldsymbol{x}, \boldsymbol{y}] \in \mathbb{R}^{d_x+d_y}$. For a fixed time interval, LQ data has the incompleteness issue, illustrated in Section 1, implying that $\mathcal{N}_y \subset \mathcal{N}_x = \{0, 1, 2, \cdots, |\mathcal{N}_x| - 1\}$, where $|\cdot|$ is the cardinality of the set. In many practical settings, $\mathcal{N}_y$ can be a small fraction of $\mathcal{N}_x$, for example, in power systems, $|\mathcal{N}_y| \approx 0.05 \times |\mathcal{N}_x|$ (Li et al., 2024b).

Our framework generates an interpolated dataset $\{\tilde{\boldsymbol{y}}(t_i)\}_{i \in \mathcal{N}_x \setminus \mathcal{N}_y}$ to align LQ data with HQ data. This process ensures that LQ data is brought up to the same standard as HQ data, significantly improving the training of $f_\Theta(s(t_i))$. By doing so, our model facilitates high-resolution predictions for LQ variables during online testing.

### 3.2 A UNIFIED GEOMETRIC REPRESENTATION

**Weight Matrix Flow-based Geometric Representation**: To capture complex and adaptive dynamics in real-time systems, we represent the parameters $\Theta(t)$ as time-dependent weight matrices. Prior work has shown that modeling the flow of a neural network's weight matrix can capture the most important parameters for learning dynamic systems (Choromanski et al., 2020b; Cho et al., 2024). Building on this idea, we propose a geometric representation that decomposes the parameter space based on HQ and LQ outputs, allowing for optimal data alignment.

$$\begin{cases} \tilde{\boldsymbol{x}}(t_i) = f_{\{\Theta_x(t_i) \cup \Theta_0\}}\big(\hat{\boldsymbol{s}}(t_{i-1})\big) \\ \tilde{\boldsymbol{y}}(t_i) = f_{\{\Theta_y(t_i) \cup \Theta_1\}}\big(\hat{\boldsymbol{s}}(t_{i-1})\big) \\ \Theta = \Theta_0 \cup \Theta_1 \cup \Theta_x \cup \Theta_y \\ \Theta_x(t_i), \Theta_y(t_i) \in \mathcal{M}, \end{cases} \tag{1}$$

Here, $\hat{s}(t_i)$ represents either the true measurements, which are a combination of HQ and LQ data $[\boldsymbol{x}(t_i), \boldsymbol{y}(t_i)]$ ($\forall i \in \mathcal{N}_y$), or a combination of HQ measurements and interpolated LQ data $[\boldsymbol{x}(t_i), \tilde{\boldsymbol{y}}(t_i)]$ ($\forall i \in \mathcal{N}_x \setminus \mathcal{N}_y$). The weight matrices $\Theta_x(t)$ and $\Theta_y(t)$ represent the dynamics of HQ and LQ data, respectively, and are modeled within a shared manifold $\mathcal{M}$. $\Theta_0$ and $\Theta_1$ are static subsets of $\Theta$, e.g., bias vectors. By decomposing the parameter space, we can explore correlations between HQ and LQ representations, leading to more accurate predictions. We make the following assumption to guide the alignment process:

**Assumption 1.** *Assume a system with low nonlinearity and limited measurement noise. The HQ and LQ states of the system, $\boldsymbol{x}(t)$ and $\boldsymbol{y}(t)$, exhibit high similarity. Therefore, the flows of $\Theta_x(t)$ and $\Theta_y(t)$ share the same shape but occupy different locations on the manifold $\mathcal{M}$.*

Assumption 1 is valid under data incompleteness and when there is limited random noise that may destroy similarity. Then, we hypothesize that the similarity can be geometrically interpreted as the same shape of $\Theta_x(t)$ and $\Theta_y(t)$. Numerically, section 4.2 illustrates that aligned shape can help G-AlignNet effectively capture the similarity and make accurate predictions.

**Geometric Optimization for Optimal Data Quality Alignment**. A key challenge in aligning HQ and LQ data is matching their underlying geometric structure. This "shape matching" ensures that the parameters associated with HQ data can guide the alignment of LQ data, allowing for accurate interpolation. To formalize this, we define a shape-matching optimization problem:

$$Q^* = \underset{Q^\top Q=I,\ \Theta_x(t),\Theta_y(t)\in\mathcal{M}}{\arg\min} \frac{1}{|\mathcal{N}_y|} \sum_{i\in\mathcal{N}_y} \left|\left|\Theta_y(t_i) - \Theta_x(t_i)Q\right|\right|_F^2, \tag{2}$$

where the goal, by Proposition 1, is to find an orthogonal matrix $Q^* \in \mathbb{R}^{n\times n}$ that best aligns HQ flow $\Theta_x(t_i)$ with LQ flow $\Theta_y(t_i)$. The Frobenius norm $||\cdot||_F$ is used to measure the difference between the aligned matrices.

**Proposition 1** (Shape Preservation via Orthogonal Matrix). *If $Q \in \mathbb{R}^{n\times n}$ is an orthogonal matrix, the flow between $\Theta_x(t)$ and $\Theta_x(t)Q$ is the same.*

This property is crucial because orthogonal transformations preserve the Euclidean norm, ensuring that the relative positions of points in the parameter space remain unchanged. It guarantees that the length and angle between any two points on the flow are preserved, as shown in Appendix C.1.

Figure 1 illustrates how HQ and LQ parameters are aligned. The dark blue and green points represent the parameters $\Theta_x(t_i)$ and $\Theta_y(t_i)$ at times $t_i \in \mathcal{N}_y$, where we have both HQ and LQ measurements. These points can be trained effectively. However, for times $t_i \in \mathcal{N}_x \setminus \mathcal{N}_y$, represented by the light green points, we rely on the shape-preserving transformation to interpolate the LQ parameters. The optimal solution $Q^*$ generates interpolated parameters $\{\tilde{\Theta}_y(t_i)\}_{i\in\mathcal{N}_x\setminus\mathcal{N}_y}$, resulting in interpolated states $\{\tilde{\boldsymbol{y}}(t_i)\}_{i\in\mathcal{N}_x\setminus\mathcal{N}_y}$.

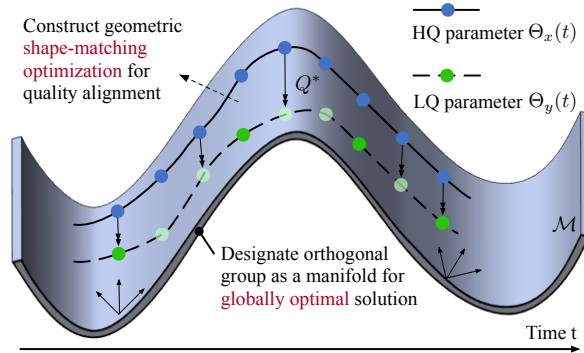

Figure 1: A unified geometric perspective for quality alignment and dynamical modeling.

This approach avoids the need to approximate complex data manifolds and instead focuses on learning within a simpler parameter manifold. For example, in Section 3.3, we will show that parameter flows are easy to learn with limited approximation error, and the interpolation error has fast convergence. To ensure that the interpolation process achieves global optimality, we need to select an appropriate manifold $\mathcal{M}$. We show that the orthogonal group $\mathcal{O}(n) = \{W \in \mathbb{R}^{n\times n} | W^\top W = I_n\}$, where $I_n$ is the identity matrix, provides the best solution.

**Proposition 2** (Globally Optimal Solution). *Suppose matrices $\Theta_x(t), \Theta_y(t) \in \mathcal{O}(n)$. The optimization problem in (2) has a global minimizer $Q^* = UV^\top$, where $\frac{1}{|\mathcal{N}_y|}\sum_{i\in\mathcal{N}_y}\Theta_x(t_i)^\top\Theta_y(t_i) = U\Sigma V^\top$ is the Singular Value Decomposition.*

This result, proven in Appendix C.2, shows that we can achieve a globally optimal alignment by solving the shape-matching problem. Furthermore, to maintain the orthogonality of $\Theta_x(t)$ and $\Theta_y(t)$ throughout the learning process, we analyze the evolution of $\Theta_x(t)$. Ensuring orthogonality is crucial because it preserves the geometric structure of the parameter space, which is necessary for accurate quality alignment. Specifically, we derive the following ODE by differentiating the orthogonality condition $\Theta_x^\top \Theta_x = I_n$:

$$\dot{\Theta}_x(t) = \Theta_x(t)\Omega_x(t), \tag{3}$$

where $\Omega_x(t)$ is a skew-symmetric matrix. As shown in (Choromanski et al., 2020b), this ODE ensures that $\Theta_x(t)$ remains within the orthogonal group $\mathcal{O}(n)$ during the learning. To model this dynamic evolution, we use a Neural ODE (Chen et al., 2018), which generates the orthogonal matrix flow via a neural network that outputs skew-symmetric matrices. Specifically, we define the neural network as: $g_{\Psi_x}(t) = \sum_i a_i\big(g_{\Psi_x^{(i)}}(t) - g_{\Psi_x^{(i)}}^\top(t)\big)$, where $a_i$ are learnable coefficients, and $g_{\Psi_x^{(j)}}(\cdot) : \mathbb{R} \to \mathbb{R}^{n \times n}$ are sub-neural networks that output random matrices. The neural network $g_{\Psi_x}(t)$ is always skew-symmetric, ensuring that the flow $\Theta_x(t)$ remains in $\mathcal{O}(n)$. This guarantees that:

$$\Theta_x(t) = \text{ODESolve}(\Theta_x(t_0), g_{\Psi_x}\Theta_x(t), t) = \Theta_x(t_0) + \int_{t_0}^{t} g_{\Psi_x}(t)\Theta_x(t)dt, \tag{4}$$

where an ODE solver is used to compute the integral over time. By generating orthogonal matrix flows through the Neural ODE, we ensure that the quality alignment between HQ and LQ data can be maintained consistently throughout the learning process.

**Zero-error Shape Matching with Global Optimality**. To further improve the alignment process, we propose the following corollary, based on Proposition 2, which guarantees global optimality with zero error under specific conditions.

**Corollary 1** (Zero-error Shape Matching). *Suppose the skew-symmetric matrix* $\Omega(t) \equiv \Omega_y(t) \equiv \Omega_x(t)$, *as defined in Equation (3). The optimization problem in Equation (2) has a unique global minimizer* $Q^*$, *and the corresponding objective value is zero.*

This result, proven in Appendix C.3, highlights that when $\Omega_x(t)$ and $\Omega_y(t)$ evolve in the same way, the shape-matching optimization achieves zero error. This occurs because the flow of both $\Theta_x(t)$ and $\Theta_y(t)$ remains perfectly aligned, leading to an optimal solution.

**Achieving Built-in Optimal Quality Alignment via Architecture Design**. Corollary 1 suggests that by using a single Neural ODE, parameterized by $\Psi$ (i.e., $\Psi \equiv \Psi_x \equiv \Psi_y$), we can generate orthogonal matrix flows for both $\Theta_x$ and $\Theta_y$. The Neural ODE ensures that the parameter flows are aligned in shape, even if their starting points differ. As long as $\Theta_x(t_0) \neq \Theta_y(t_0)$, their flows will occupy different locations on $\mathcal{O}(n)$, preserving their relative alignment.

We refer to this architecture as **G-AlignNet**: a *Quality-Aligned Geometric Network*. The architecture is designed to search for two parameter flows with the same shape, optimizing $\Psi$ to best fit the output data at times $\{t_i\}_{i \in \mathcal{N}_y}$ and implicitly solving the shape-matching optimization in Equation (2). This allows us to analyze the error between the two flows in terms of shape differences, as discussed in Section 3.3. Figure 2 illustrates the G-AlignNet architecture, where the base model can vary depending on the target system, such as RNNs or INRs.

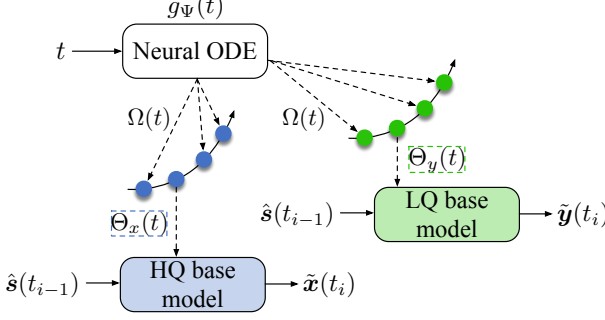

Figure 2: The framework of the proposed G-AlignNet.

**Shaping the RNN Weight Flow**. For Recurrent Neural Networks (RNNs), we design time-dependent cells to incorporate the adaptive behavior of the system. Let $\boldsymbol{h}_i \in \mathbb{R}^n$ denote the hidden state at time $t_i$. The RNN cell is defined as:

$$\boldsymbol{h}_i = \sigma\big(\Theta_x(t_i)\boldsymbol{h}_{i-1} + W\hat{\boldsymbol{s}}(t_{i-1}) + \boldsymbol{b}\big), \tag{5}$$

where $\sigma(\cdot)$ is the nonlinear activation function, $\Theta_x(t_i)$ and $W$ are weight matrices for hidden states and inputs, respectively, and $\boldsymbol{b}$ is a bias term. The matrix $\Theta_x(t)$ processes the temporal dependencies in $\boldsymbol{h}_{i-1}$, continuously adapting the model to evolving systems.

**Shaping the INR Weight Flow**. For Implicit Neural Representations (INRs) (Yin et al., 2022) and PINNs (Cho et al., 2024), the core is a fully-connected neural network that processes hidden features $\boldsymbol{h}_i$ (Fathony et al., 2020; Sitzmann et al., 2020). We propose the following time-dependent version:

$$\boldsymbol{h}_1 = \sigma(Wt + \boldsymbol{b}_1), \boldsymbol{h}_j = \sigma(\Theta_x^{(j)}(t)\boldsymbol{h}_{j-1} + \boldsymbol{b}_j), \tag{6}$$

where time $t$ is the input for predicting values of $\boldsymbol{x}(t)$, $j$ is the layer index, and $\{\Theta_x^{(j)}(t)\}_j$ are the time-dependent weight flows for each layer.

**End-to-end G-AlignNet Training**. The training loss is calculated using the Mean Square Error (MSE), which can be minimized using gradient-based optimization methods. Since $\Theta_x(t)$ and $\Theta_y(t)$ are generated by the Neural ODE in Equation (4), the only parameters that require updates are $\{\Theta_0, \Theta_1, \Psi\}$, making this approach sample-efficient compared to traditional methods that require optimization of $\Theta_x(t)$ and $\Theta_y(t)$ independently.

## 3.3 THEORETICAL ANALYSIS FOR OPTIMALITY AND ERROR BOUNDS

In this section, we analyze two primary sources of error that affect the accuracy of HQ and LQ data alignment: (1) measurement noises, which causes *random distortions* in the true parameter flows, and (2) approximation error from the Neural ODE, which leads to *deviations* between the true and learned parameter flows. These error sources play a crucial role in determining the overall performance of G-AlignNet when aligning HQ and LQ data. To model these errors, we assume the following relationships:

$$\bar{\Theta}_y(t_i) = \bar{\Theta}_x(t_i)\bar{Q}(I_n + E_i), \ \bar{\Theta}_x(t_i) = \Theta_x(t_i) + D_i, \tag{7}$$

where $E_i \sim \mathcal{N}(0, \sigma_0^2 \mathbf{1})$ represents Gaussian noise for shape distortions, and $D_i \in \mathbb{R}^{n \times n}$ represents biased deviations at time $t_i$. Here, $\sigma_0$ is the standard deviation, and $\mathbf{1}$ is an all-one matrix. The first equation models the impact of shape distortion using a multiplicative error, while the second captures approximation errors in the learned flow.

**Proposition 3** (Interpolation Error Bound). *Given the error model in Equation (7), training G-AlignNet approximates the true parameters by solving the following optimization:*

$$Q^{**} = \arg \min_{Q^\top Q = I} \frac{1}{|\mathcal{N}_y|} \sum_{i \in \mathcal{N}_y} \left\| \bar{\Theta}_y(t_i) - \bar{\Theta}_x(t_i)Q \right\|_F^2. \tag{8}$$

*This leads to an estimation error for matrix $Q^{**}$, relative to the true transformation matrix $\bar{Q}$:*

$$\mathbb{E} \left\| Q^{**} - \bar{Q} \right\|_F \leq n^{\frac{3}{2}} \sigma_0 / \sqrt{|\mathcal{N}_y|}, \tag{9}$$

*and an interpolation error for the parameter $\tilde{\Theta}_y(t_i) = \Theta_x(t_i)Q^{**}, i \in \mathcal{N}_x \backslash \mathcal{N}_y$:*

$$\mathbb{E}_{i \in \mathcal{N}_x \backslash \mathcal{N}_y} \left\| \tilde{\Theta}_y(t_i) - \bar{\Theta}_y(t_i) \right\|_F \leq n^2 \sigma_0 / \sqrt{|\mathcal{N}_y|} + n^{\frac{1}{2}} \varepsilon_0, \tag{10}$$

*where $\varepsilon_0 = \frac{1}{|\mathcal{N}_x|} \sum_{i \in \mathcal{N}_x} \|D_i\|_F$ represents the average approximation error due to the Neural ODE.*

The detailed proof can be found in Appendix C.4. Intuitively, these bounds show that the estimation error for matrix $Q^{**}$ converges at a rate of $\mathcal{O}(\frac{1}{\sqrt{|\mathcal{N}_y|}})$, which is consistent with state-of-the-art results in compressed sensing (Iwen et al., 2021; Wang et al., 2017). The error bound from noise indicates that our model is robust to Gaussian noise with low variance. We need further investigations into the model's performance under high noise levels. Next, we analyze the error induced by the Neural ODE model.

**Proposition 4** (Neural ODE Approximation Error). *The average approximation error for the Neural ODE model can be bounded as follows (Hillebrecht & Unger, 2022; Soetaert et al., 2012):*

$$\varepsilon_0 \leq \mathcal{O}\left( \frac{1}{|\mathcal{N}_x|} \left\| \Theta_x(t_0) - \bar{\Theta}_x(t_0) \right\|_F \frac{1 - \beta e^{\alpha T}}{1 - \beta e^{\frac{\alpha T}{|\mathcal{N}_x|}}} + \frac{h^{p+1}}{|\mathcal{N}_x|} T e^{\alpha T/|\mathcal{N}_x|} + h^{p+1} \sum_{k=0}^{K-1} e_k^{adj} \right), \tag{11}$$

*where $T = t_{|\mathcal{N}_x|}$ is the end time for HQ data, $\beta > 1$ is a constant, and $\alpha$ is the largest signal value of all matrices $\Omega(t_i), i \in \mathcal{N}_x$. $h$ and $p$ are the step size and the order of the Neural ODE solver, respectively. For the adjoint method, $K$ is the number of discretized points in forward/reverse integration (Zhuang et al., 2020). $e_k^{adj} > 0$ represents the reverse inaccuracy factor in the adjoint method. $e_k^{adj} = 0$ in the naive method or Adaptive Checkpoint Adjoint (ACA) (Zhuang et al., 2020).*

The proof, provided in Appendix C.5, is based on analyzing the perturbed initial value problem for the ODE system (Soetaert et al., 2012). By leveraging the linearity of the ODE in Equation (3), we relate the solution's Lipschitz constant to the largest singular value $\alpha$. In our model, $\alpha$ is kept small through normalization, which ensures a faster convergence rate for the error.

Proposition 4 demonstrates that for a fixed time horizon $T$, the error convergence rate is approximately $\mathcal{O}(\frac{1}{|\mathcal{N}_x|})$, significantly outperforming manifold-based methods with convergence rates of $\mathcal{O}(\frac{1}{\log |\mathcal{N}_y|})$ (Iwen et al., 2021). This faster convergence is a result of the efficient use of HQ data in the Neural ODE, combined with the representational power of the G-AlignNet framework.

## 4 EXPERIMENTS

### 4.1 SETTINGS

G-AlignNet is applicable to diverse systems, including: (1) **Residential Electricity Consumption**. We gather real-world electricity data, public available at (Pacific Gas and Electric Company, 2024). Forecasting this data is crucial for planning in power markets (Xu et al., 2018). (2) **Photovoltaic System**. We introduce a publicly available Photovoltaic (PV) dataset (Boyd, 2016) for solar power generations, adaptive to the movement of the sun and the wind. (3) **Power System Event Measurements**. The synchrophasor measurements are sampled during a power grid transient process after a three-phase fault (Li et al., 2019), which is a high-order and time-dependent ODE system. (4) **Air Quality System**. UCI Repository provides measurements of metal oxide chemical sensors in air quality monitoring system (Vito, 2008). (5) **Synthetic 2-D Spiral Dataset**. We test a continuous ODE system to demonstrate the model's capacity to understand the true structures for extrapolations. The data generation process is in Appendix D.1. System dimensions are shown in Appendix D.4. Our test systems have moderate nonliearity and no measurement noise. However, the available data amount largely varies to create data incompleteness.

To test our systems, we consider interpolation (Sections 4.2 and 4.3), extrapolation (Section 4.3), and control tasks (Appendix D.5). They are important to evaluate the performance of the learned dynamic model in real-world systems. Moreover, we conduct sensitivity analysis in Section 4.4 to evaluate the model robustness to data quality levels, and give intuitive visualization in Section 4.5.

The following benchmark methods are used. For interpolation, we have: (1) **Linear Spline** and (2) **Cubic Spline**. This method applies linear and cubic polynomials to approximate the underling signals. (3) **Compressed Sensing**. CS explores the manifold data structure by assuming a linear format for signal recovery (Donoho, 2006). (4) **Deep CS.** DCS combines the deep generative models with CS. We utilize a Variational Autoencoder with CS to recover the data streams (Bora et al., 2017; Wu et al., 2019). (5) **Semi-supervised NN.** Semi-NN utilizes DNN to map from HQ or past LQ to current LQ data, and the semi-supervised framework facilitates to use all information (Ma et al., 2023). (6) **Multiplicative Filter Network**. MFN is a cutting-edge INR model to map from time to system states (Fathony et al., 2020). Adaptive filtering is incorporated to make the model capable of handling time-evolving systems.

For dynamic prediction, we employ: (1) **Recurrent Neural Network**. RNN sequentially process data with hidden cells to store past information. (2) **ODE-RNN**. In ODE-RNN, Neural ODE is embedded to learn the dynamical function of hidden states between every two arbitrary timestamps (Rubanova et al., 2019). (3) **Neural Controlled Differential Equation**. Neural CDE creates a continuous data path to control the evolution of the state's ODE flow, suitable for irregularly sampled data (Kidger et al., 2020). (4) **MFN**. Described above. (5) **Neural ODE + RNN** and (6) **Neural ODE + MFN**. Similar to our design, Neural ODE can be used to as a hyper-network to control the flow of RNNs and MFNs. This brings additional adaptivity to the base model (Yin et al., 2022). As comparisons, we don't restrict the shape of ODE flows. *In general, discrete sequence model (RNN), continuous models (ODE-RNN, Neural CDEm MFN), and parameter flow-based models*

*(Neural ODE + RNN/Neural ODE + MFN) are comprehensively utilized.* We use G-AlignNetR and G-AlignNetI to represent the model with RNN and INR (i.e., MFN) as the base model, respectively. To evaluate all the methods, we calculate the Mean Absolute Percentage Error (MAPE(%)) and Mean Square Error (MSE) between the interpolated/forecast and the true measurements.

### 4.2 EXACT SHAPE-MATCHING TO OPTIMALLY ALIGN DATA QUALITIES

We first validate the effectiveness of shape-matching for $\Theta_x(t)$ and $\Theta_y(t)$ by comparing G-AlignNetR and Neural ODE + RNN (i.e., no restriction on the RNN parameter flow). These models are trained on high-resolution (HR) and low-resolution (LR) load data with the data coverage rate to be $2.5\%$. To visualize the two flows, we reduce the dimensionality with Principal Component Analysis (PCA). We also do centralization for the flows to exhibit the shape difference. To make sure that the two flows have an orthogonal relation, we utilize a checking program in Section D.2. In each iteration, the orthogonality error is around $10^{-8}$.

Figure 3 illustrates the 3-D plot of parameter flows. As shown in the left part, G-AlignNetR can achieve a perfect match for $\Theta_x(t)$ and $\Theta_y(t)$, but Neural ODE + RNN can't. As a result, with limited LR data in green points, G-AlignNetR has much better prediction results for LR dynamics (green curves in the right part).

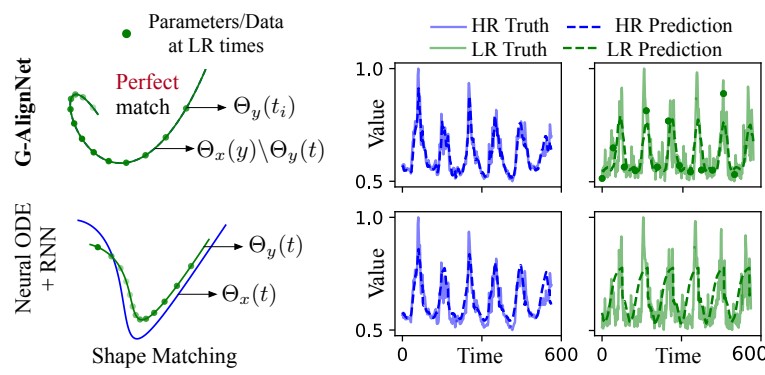

Figure 3: Exact shape matching leads to better LR predictions.

This observation implies that naive HR-LR information fusion in Neural ODE + RNN is insufficient. Instead, we use shape matching to fully exploit HR/LR data similarity (i.e., the blue and green solid curves). Hence, in our model, HR provides much better guidance for LR data interpolation.

### 4.3 QUALITY ALIGNMENT BOOSTS PERFORMANCES UNDER DIFFERENT QUALITY ISSUES

We evaluate the model performance for LR data (90% data drop rate with a fixed interval), missing observations (20% data drops with consecutive intervals), and irregularly sampled data (30% random data drop rate). We present the complete results in interpolation and extrapolation tasks in Table 1 and 2, respectively. The optimal quality alignment in G-AlignNetR brings superior performance for most scenarios. For interpolation, G-AlignNetR gains big improvements for power event and spiral data with high oscillations. However, such complex dynamics prevent other interpolation methods. In general, spline and DCS methods can achieve comparable results. Spline works when the data drop rate is low, and DCS works when HQ and LQ data have the same distribution. For instance, in photovoltaic (PV) and air quality systems, both HQ and LQ data are typically collected within the same local region under consistent weather conditions, resulting in similar measurements for solar generation and air quality. These are special cases when there is a high chance of easily understanding the data structure. G-AlignNetR doesn't need this requirement and is generally applicable due to our efficient abstraction of common knowledge, i.e., the shape of the parameter flow. Moreover, such knowledge is optimally aligned using our geometric optimization.

For fair comparisons, all extrapolation benchmark methods incorporate the interpolated data from Cubic Spline, which brings stable and comparable interpolation results in several cases. G-AlignNet performs best in all test cases. Specifically, RNN as a discrete sequence model is competitive when the interpolated data is good. MFN doesn't work well because solely inputting time is insufficient for test system states. ODE-RNN and Neural CDE have comparable results for load and air quality data that have cyclic patterns and a relatively simple data structure. Neural ODE + RNN/MFN as flow-based models can also work well for power event data whose magnitudes are gradually damped to 0. However, although these two methods have a high adaptation capacity to evolving dynamics, they perform badly in another adaptive system, i.e., spiral, because their input interpolated data have errors. Our G-AlignNetR, however, tackles complex structures with the best data quality alignment.

G-AlignNetI works well in continuous systems like power events, air quality, and spiral datasets and achieves state-of-the-art performance with around $1\% \sim 10\%$ error reduction compared to G-AlignNetR methods. However, for systems with more uncertainty, e.g., the load and PV systems, G-AlignNetI's performance is not competitive. The main reason is that the INR model is less powerful than RNN in capturing historical trends and patterns for predictions.

Table 1: Performance (mean ± standard deviation) of the Interpolation Tasks.

| Data | Scenario | Metric | Linear Spline | Cubic Spline | CS | DCS | Semi-NN | MFN | G-AlignNetR | G-AlignNetI |
|---|---|---|---|---|---|---|---|---|---|---|
| Load | LR | MSE ($10^{-2}$) | $1.53 \pm 0.23$ | $1.70 \pm 0.26$ | $2.02 \pm 0.30$ | $2.96 \pm 0.44$ | $1.37 \pm 0.21$ | $2.47 \pm 0.37$ | $\mathbf{1.10 \pm 0.16}$ | $1.96 \pm 0.37$ |
| | | MAPE (%) | $14.71 \pm 1.87$ | $15.57 \pm 2.03$ | $16.55 \pm 1.80$ | $24.68 \pm 3.00$ | $14.56 \pm 2.18$ | $17.99 \pm 2.16$ | $\mathbf{13.16 \pm 1.47}$ | $16.23 \pm 1.88$ |
| | Missing | MSE ($10^{-2}$) | $0.80 \pm 0.12$ | $1.14 \pm 0.17$ | $1.34 \pm 0.20$ | $1.79 \pm 0.27$ | $1.07 \pm 0.16$ | $1.80 \pm 0.27$ | $\mathbf{0.72 \pm 0.11}$ | $1.05 \pm 0.27$ |
| | | MAPE (%) | $8.75 \pm 1.06$ | $12.05 \pm 1.62$ | $12.83 \pm 1.40$ | $16.34 \pm 1.87$ | $9.37 \pm 1.34$ | $12.57 \pm 1.82$ | $\mathbf{7.63 \pm 0.71}$ | $10.98 \pm 1.12$ |
| | Irregular | MSE ($10^{-2}$) | $1.24 \pm 0.19$ | $1.70 \pm 0.25$ | $2.06 \pm 0.31$ | $2.96 \pm 0.44$ | $1.37 \pm 0.21$ | $2.47 \pm 0.37$ | $\mathbf{1.10 \pm 0.17}$ | $1.22 \pm 0.26$ |
| | | MAPE (%) | $13.21 \pm 1.89$ | $13.71 \pm 1.52$ | $15.48 \pm 1.96$ | $19.86 \pm 2.95$ | $14.37 \pm 1.68$ | $15.41 \pm 1.69$ | $\mathbf{12.38 \pm 1.60}$ | $13.31 \pm 1.89$ |
| PV | LR | MSE ($10^{-2}$) | $0.54 \pm 0.08$ | $0.70 \pm 0.11$ | $1.37 \pm 0.20$ | $\mathbf{0.40 \pm 0.10}$ | $1.86 \pm 0.28$ | $2.21 \pm 0.63$ | $0.47 \pm 0.07$ | $0.58 \pm 0.13$ |
| | | MAPE (%) | $8.52 \pm 1.28$ | $7.95 \pm 1.07$ | $14.19 \pm 1.65$ | $\mathbf{6.52 \pm 0.78}$ | $15.06 \pm 2.06$ | $22.89 \pm 3.03$ | $7.39 \pm 0.91$ | $8.56 \pm 1.32$ |
| | Missing | MSE ($10^{-2}$) | $0.24 \pm 0.04$ | $0.33 \pm 0.05$ | $0.69 \pm 0.10$ | $0.28 \pm 0.03$ | $0.93 \pm 0.14$ | $1.81 \pm 0.32$ | $\mathbf{0.22 \pm 0.03}$ | $0.24 \pm 0.05$ |
| | | MAPE (%) | $5.80 \pm 0.75$ | $5.80 \pm 0.87$ | $11.65 \pm 1.40$ | $5.19 \pm 0.58$ | $12.54 \pm 1.63$ | $17.67 \pm 2.65$ | $\mathbf{5.11 \pm 0.73}$ | $5.71 \pm 0.79$ |
| | Irregular | MSE ($10^{-2}$) | $0.35 \pm 0.05$ | $0.44 \pm 0.07$ | $0.89 \pm 0.13$ | $0.45 \pm 0.03$ | $1.40 \pm 0.21$ | $0.86 \pm 0.47$ | $\mathbf{0.31 \pm 0.05}$ | $0.34 \pm 0.08$ |
| | | MAPE (%) | $7.00 \pm 0.98$ | $7.00 \pm 1.05$ | $13.00 \pm 1.55$ | $8.92 \pm 1.14$ | $14.25 \pm 2.00$ | $9.78 \pm 2.85$ | $\mathbf{6.75 \pm 0.91}$ | $6.87 \pm 0.98$ |
| Power event | LR | MSE ($10^{-2}$) | $0.38 \pm 0.05$ | $0.27 \pm 0.04$ | $0.61 \pm 0.09$ | $0.51 \pm 0.08$ | $0.70 \pm 0.11$ | $1.17 \pm 0.18$ | $\mathbf{0.07 \pm 0.01}$ | $\mathbf{0.07 \pm 0.01}$ |
| | | MAPE (%) | $5.29 \pm 0.31$ | $4.39 \pm 0.32$ | $7.84 \pm 1.02$ | $7.94 \pm 1.19$ | $7.85 \pm 1.18$ | $11.37 \pm 1.64$ | $3.21 \pm 0.38$ | $\mathbf{3.15 \pm 0.35}$ |
| | Missing | MSE ($10^{-2}$) | $0.04 \pm 0.01$ | $0.03 \pm 0.04$ | $0.31 \pm 0.05$ | $0.26 \pm 0.04$ | $0.35 \pm 0.05$ | $0.59 \pm 0.09$ | $0.05 \pm 0.01$ | $\mathbf{0.03 \pm 0.01}$ |
| | | MAPE (%) | $2.20 \pm 0.30$ | $2.03 \pm 0.28$ | $5.94 \pm 0.89$ | $5.35 \pm 0.80$ | $6.27 \pm 0.94$ | $9.45 \pm 1.41$ | $1.87 \pm 0.28$ | $\mathbf{1.14 \pm 0.11}$ |
| | Irregular | MSE ($10^{-2}$) | $0.06 \pm 0.01$ | $\mathbf{0.07 \pm 0.01}$ | $0.39 \pm 0.06$ | $0.33 \pm 0.05$ | $0.45 \pm 0.07$ | $0.77 \pm 0.12$ | $0.08 \pm 0.02$ | $\mathbf{0.07 \pm 0.01}$ |
| | | MAPE (%) | $3.01 \pm 0.45$ | $2.85 \pm 0.43$ | $6.72 \pm 1.01$ | $6.20 \pm 0.93$ | $6.97 \pm 1.05$ | $10.33 \pm 1.55$ | $\mathbf{2.75 \pm 0.41}$ | $2.70 \pm 0.40$ |
| Air quality | LR | MSE ($10^{-2}$) | $0.83 \pm 0.10$ | $0.96 \pm 0.12$ | $1.04 \pm 0.13$ | $0.70 \pm 0.10$ | $0.92 \pm 0.11$ | $0.95 \pm 0.20$ | $0.70 \pm 0.08$ | $\mathbf{0.65 \pm 0.05}$ |
| | | MAPE (%) | $10.02 \pm 1.33$ | $10.71 \pm 1.17$ | $12.98 \pm 1.81$ | $\mathbf{9.04 \pm 1.02}$ | $11.73 \pm 1.58$ | $10.51 \pm 1.26$ | $9.87 \pm 1.21$ | $9.04 \pm 1.04$ |
| | Missing | MSE ($10^{-2}$) | $0.57 \pm 0.07$ | $0.77 \pm 0.10$ | $0.65 \pm 0.08$ | $\mathbf{0.4 \pm 0.10}$ | $0.71 \pm 0.09$ | $0.83 \pm 0.40$ | $0.50 \pm 0.06$ | $0.41 \pm 0.09$ |
| | | MAPE (%) | $5.76 \pm 0.58$ | $8.34 \pm 1.22$ | $9.01 \pm 1.19$ | $\mathbf{5.21 \pm 0.74}$ | $7.71 \pm 1.06$ | $10.30 \pm 1.67$ | $5.31 \pm 0.67$ | $5.28 \pm 0.70$ |
| | Irregular | MSE ($10^{-2}$) | $0.79 \pm 0.10$ | $0.92 \pm 0.12$ | $0.84 \pm 0.10$ | $0.85 \pm 0.16$ | $0.79 \pm 0.10$ | $1.07 \pm 0.44$ | $0.61 \pm 0.07$ | $\mathbf{0.58 \pm 0.06}$ |
| | | MAPE (%) | $9.95 \pm 1.22$ | $9.16 \pm 0.99$ | $10.57 \pm 1.26$ | $10.03 \pm 1.14$ | $9.57 \pm 1.43$ | $11.61 \pm 1.09$ | $9.14 \pm 1.03$ | $\mathbf{8.39 \pm 1.00}$ |
| Spiral | LR | MSE ($10^{-2}$) | $1.11 \pm 0.14$ | $1.57 \pm 0.21$ | $4.64 \pm 0.58$ | $3.46 \pm 0.45$ | $1.85 \pm 0.24$ | $2.65 \pm 0.30$ | $\mathbf{0.15 \pm 0.02}$ | $0.18 \pm 0.02$ |
| | | MAPE (%) | $7.48 \pm 0.75$ | $8.65 \pm 1.21$ | $15.49 \pm 1.76$ | $15.03 \pm 2.04$ | $11.31 \pm 1.13$ | $14.68 \pm 2.35$ | $2.97 \pm 0.35$ | $\mathbf{2.95 \pm 0.35}$ |
| | Missing | MSE ($10^{-2}$) | $0.75 \pm 0.10$ | $1.00 \pm 0.13$ | $3.26 \pm 0.42$ | $2.46 \pm 0.32$ | $1.25 \pm 0.16$ | $1.91 \pm 0.20$ | $0.10 \pm 0.01$ | $\mathbf{0.08 \pm 0.01}$ |
| | | MAPE (%) | $4.77 \pm 0.57$ | $5.87 \pm 0.63$ | $7.96 \pm 0.89$ | $7.78 \pm 1.14$ | $8.63 \pm 1.15$ | $6.19 \pm 0.86$ | $1.71 \pm 0.25$ | $\mathbf{1.65 \pm 0.21}$ |
| | Irregular | MSE ($10^{-2}$) | $1.07 \pm 0.14$ | $1.46 \pm 0.19$ | $3.83 \pm 0.50$ | $3.11 \pm 0.40$ | $1.56 \pm 0.20$ | $1.72 \pm 0.20$ | $\mathbf{0.13 \pm 0.02}$ | $\mathbf{0.13 \pm 0.01}$ |
| | | MAPE (%) | $6.59 \pm 0.95$ | $8.11 \pm 1.16$ | $15.13 \pm 1.53$ | $14.17 \pm 1.79$ | $11.11 \pm 1.27$ | $9.25 \pm 1.09$ | $2.83 \pm 0.31$ | $\mathbf{2.55 \pm 0.28}$ |

Table 2: Performance (mean ± standard deviation) of the Extrapolation Tasks.

| Data | Scenario | Metric | RNN | ODE-RNN | Neural CDE | MFN | NODE + RNN | NODE + MFN | G-AlignNetR | G-AlignNetI |
|---|---|---|---|---|---|---|---|---|---|---|
| Load | LR | MSE ($10^{-2}$) | $1.33 \pm 0.16$ | $1.31 \pm 0.15$ | $1.43 \pm 0.19$ | $2.61 \pm 0.31$ | $1.32 \pm 0.17$ | $1.60 \pm 0.16$ | $\mathbf{1.00 \pm 0.13}$ | $1.41 \pm 0.15$ |
| | | MAPE (%) | $13.29 \pm 1.88$ | $13.34 \pm 1.57$ | $13.70 \pm 1.83$ | $20.73 \pm 2.64$ | $14.96 \pm 2.09$ | $15.32 \pm 1.83$ | $\mathbf{12.47 \pm 1.40}$ | $13.54 \pm 1.48$ |
| | Missing | MSE ($10^{-2}$) | $1.02 \pm 0.14$ | $1.01 \pm 0.12$ | $0.78 \pm 0.10$ | $1.67 \pm 0.20$ | $0.93 \pm 0.13$ | $0.82 \pm 0.12$ | $\mathbf{0.64 \pm 0.07}$ | $0.72 \pm 0.07$ |
| | | MAPE (%) | $10.58 \pm 1.05$ | $8.45 \pm 0.91$ | $8.30 \pm 0.91$ | $14.39 \pm 1.92$ | $10.54 \pm 1.08$ | $11.65 \pm 1.32$ | $\mathbf{7.96 \pm 1.08}$ | $10.63 \pm 1.25$ |
| | Irregular | MSE ($10^{-2}$) | $1.17 \pm 0.14$ | $1.26 \pm 0.16$ | $1.31 \pm 0.14$ | $2.49 \pm 0.26$ | $1.06 \pm 0.14$ | $1.56 \pm 0.22$ | $\mathbf{0.86 \pm 0.09}$ | $1.44 \pm 0.21$ |
| | | MAPE (%) | $12.61 \pm 1.88$ | $12.20 \pm 1.29$ | $11.70 \pm 1.47$ | $16.90 \pm 1.84$ | $14.91 \pm 2.11$ | $15.13 \pm 1.91$ | $\mathbf{11.64 \pm 1.18}$ | $14.98 \pm 2.01$ |
| PV | LR | MSE ($10^{-2}$) | $0.88 \pm 0.09$ | $1.13 \pm 0.15$ | $1.34 \pm 0.18$ | $2.88 \pm 0.37$ | $2.58 \pm 0.37$ | $2.86 \pm 0.34$ | $\mathbf{0.49 \pm 0.06}$ | $1.12 \pm 0.19$ |
| | | MAPE (%) | $11.21 \pm 1.39$ | $12.09 \pm 1.26$ | $12.09 \pm 1.33$ | $21.94 \pm 6.02$ | $22.75 \pm 3.15$ | $24.59 \pm 3.55$ | $\mathbf{7.27 \pm 0.85}$ | $12.35 \pm 1.28$ |
| | Missing | MSE ($10^{-2}$) | $0.70 \pm 0.08$ | $0.70 \pm 0.10$ | $1.00 \pm 0.11$ | $1.59 \pm 0.21$ | $1.85 \pm 0.20$ | $2.11 \pm 0.22$ | $\mathbf{0.30 \pm 0.04}$ | $0.54 \pm 0.06$ |
| | | MAPE (%) | $7.23 \pm 0.82$ | $9.44 \pm 0.94$ | $6.59 \pm 0.83$ | $12.27 \pm 0.96$ | $15.12 \pm 1.72$ | $14.91 \pm 1.94$ | $\mathbf{4.20 \pm 0.47}$ | $6.3 \pm 1.02$ |
| | Irregular | MSE ($10^{-2}$) | $0.82 \pm 0.10$ | $1.09 \pm 0.15$ | $1.18 \pm 0.13$ | $1.76 \pm 0.38$ | $2.07 \pm 0.24$ | $2.77 \pm 0.33$ | $\mathbf{0.41 \pm 0.05}$ | $0.84 \pm 0.41$ |
| | | MAPE (%) | $9.61 \pm 1.23$ | $11.53 \pm 1.16$ | $11.96 \pm 1.36$ | $12.69 \pm 1.46$ | $18.73 \pm 2.73$ | $21.47 \pm 3.00$ | $\mathbf{5.97 \pm 0.88}$ | $9.88 \pm 1.35$ |
| Power event | LR | MSE ($10^{-2}$) | $0.89 \pm 0.14$ | $0.84 \pm 0.03$ | $0.91 \pm 0.05$ | $1.51 \pm 0.26$ | $0.75 \pm 0.13$ | $0.61 \pm 0.08$ | $0.29 \pm 0.04$ | $\mathbf{0.25 \pm 0.03}$ |
| | | MAPE (%) | $6.57 \pm 0.65$ | $5.89 \pm 0.51$ | $6.80 \pm 0.78$ | $10.56 \pm 1.02$ | $6.96 \pm 0.73$ | $5.32 \pm 0.54$ | $3.47 \pm 0.49$ | $\mathbf{3.15 \pm 0.43}$ |
| | Missing | MSE ($10^{-2}$) | $0.25 \pm 0.04$ | $0.23 \pm 0.01$ | $0.20 \pm 0.02$ | $1.72 \pm 0.24$ | $0.19 \pm 0.02$ | $0.22 \pm 0.02$ | $\mathbf{0.17 \pm 0.02}$ | $0.18 \pm 0.03$ |
| | | MAPE (%) | $3.18 \pm 0.41$ | $3.31 \pm 0.48$ | $4.62 \pm 0.54$ | $7.88 \pm 0.86$ | $3.34 \pm 0.48$ | $3.33 \pm 0.38$ | $\mathbf{2.31 \pm 0.24}$ | $2.35 \pm 0.24$ |
| | Irregular | MSE ($10^{-2}$) | $0.48 \pm 0.03$ | $0.43 \pm 0.03$ | $0.48 \pm 0.04$ | $0.78 \pm 0.05$ | $0.33 \pm 0.03$ | $0.36 \pm 0.04$ | $0.19 \pm 0.04$ | $\mathbf{0.13 \pm 0.03}$ |
| | | MAPE (%) | $4.81 \pm 0.52$ | $4.16 \pm 0.46$ | $4.66 \pm 0.69$ | $13.39 \pm 1.40$ | $4.00 \pm 0.60$ | $5.01 \pm 0.57$ | $2.79 \pm 0.37$ | $\mathbf{2.53 \pm 0.30}$ |
| Air quality | LR | MSE ($10^{-2}$) | $0.58 \pm 0.08$ | $0.62 \pm 0.09$ | $0.48 \pm 0.06$ | $1.03 \pm 0.26$ | $0.74 \pm 0.10$ | $0.94 \pm 0.12$ | $\mathbf{0.38 \pm 0.10}$ | $0.40 \pm 0.13$ |
| | | MAPE (%) | $10.81 \pm 1.51$ | $10.70 \pm 1.32$ | $10.18 \pm 1.27$ | $15.01 \pm 1.83$ | $12.52 \pm 1.76$ | $13.56 \pm 1.84$ | $\mathbf{6.86 \pm 0.85}$ | $7.17 \pm 0.96$ |
| | Missing | MSE ($10^{-2}$) | $0.62 \pm 0.05$ | $0.79 \pm 0.06$ | $0.66 \pm 0.07$ | $0.81 \pm 0.07$ | $0.64 \pm 0.07$ | $0.64 \pm 0.08$ | $0.51 \pm 0.06$ | $\mathbf{0.50 \pm 0.06}$ |
| | | MAPE (%) | $10.61 \pm 1.02$ | $11.35 \pm 1.07$ | $9.02 \pm 1.35$ | $11.32 \pm 1.48$ | $10.71 \pm 1.52$ | $11.65 \pm 1.79$ | $8.45 \pm 0.92$ | $\mathbf{8.31 \pm 0.91}$ |
| | Irregular | MSE ($10^{-2}$) | $0.77 \pm 0.17$ | $0.82 \pm 0.08$ | $0.73 \pm 0.05$ | $0.72 \pm 0.12$ | $0.68 \pm 0.09$ | $0.65 \pm 0.11$ | $0.50 \pm 0.07$ | $\mathbf{0.34 \pm 0.04}$ |
| | | MAPE (%) | $11.75 \pm 1.23$ | $12.25 \pm 1.46$ | $11.33 \pm 1.54$ | $11.23 \pm 1.84$ | $11.59 \pm 1.55$ | $10.85 \pm 1.76$ | $8.69 \pm 1.11$ | $\mathbf{6.52 \pm 0.89}$ |
| Spiral | LR | MSE ($10^{-2}$) | $1.21 \pm 0.15$ | $3.07 \pm 0.39$ | $1.12 \pm 0.14$ | $2.32 \pm 0.57$ | $1.02 \pm 0.14$ | $1.10 \pm 0.39$ | $\mathbf{0.19 \pm 0.02}$ | $0.21 \pm 0.03$ |
| | | MAPE (%) | $10.43 \pm 1.35$ | $17.17 \pm 2.32$ | $8.84 \pm 1.10$ | $15.12 \pm 3.43$ | $10.11 \pm 1.56$ | $10.29 \pm 1.67$ | $\mathbf{4.70 \pm 0.68}$ | $4.88 \pm 0.72$ |
| | Missing | MSE ($10^{-2}$) | $0.79 \pm 0.11$ | $1.84 \pm 0.23$ | $0.91 \pm 0.12$ | $0.83 \pm 0.17$ | $0.89 \pm 0.14$ | $0.95 \pm 0.27$ | $\mathbf{0.14 \pm 0.02}$ | $\mathbf{0.14 \pm 0.02}$ |
| | | MAPE (%) | $6.12 \pm 0.82$ | $9.55 \pm 1.22$ | $7.19 \pm 1.02$ | $8.77 \pm 1.56$ | $7.03 \pm 1.46$ | $8.25 \pm 1.73$ | $\mathbf{3.56 \pm 0.51}$ | $3.88 \pm 0.53$ |
| | Irregular | MSE ($10^{-2}$) | $1.03 \pm 0.14$ | $2.56 \pm 0.33$ | $1.09 \pm 0.13$ | $1.69 \pm 0.29$ | $1.75 \pm 0.36$ | $1.89 \pm 0.38$ | $\mathbf{0.16 \pm 0.02}$ | $0.21 \pm 0.04$ |
| | | MAPE (%) | $8.24 \pm 1.10$ | $12.75 \pm 1.89$ | $8.13 \pm 1.23$ | $9.67 \pm 1.54$ | $10.29 \pm 1.87$ | $11.79 \pm 2.08$ | $\mathbf{4.09 \pm 0.57}$ | $4.77 \pm 0.89$ |

### 4.4 SENSITIVITY ANALYSIS: LIMITED RESOLUTIONS ARE USEFUL

In this subsection, we conduct a sensitivity analysis using the Load datasets and vary different LQ data coverage rates. The interpolation and extrapolation results are in the left and the right part of Figure 4. As the rate increases, all methods consistently improve their performances for interpolation. Cubic and linear splines have almost the same error-decreasing ratio as G-AlignNet, about $\mathcal{O}(\frac{1}{|\mathcal{N}_y|})$ in Equation (10). CS and DCS's convergence rate is higher because they have additional manifold approximation errors for the data manifold, which is around $\mathcal{O}(\frac{1}{\log |\mathcal{N}_y|})$. For extrapolation, although all baselines have improvements with more data, there is still a gap between their results to G-AlignNet, which suggests that $20\%$ is still insufficient for tackling LQ data.

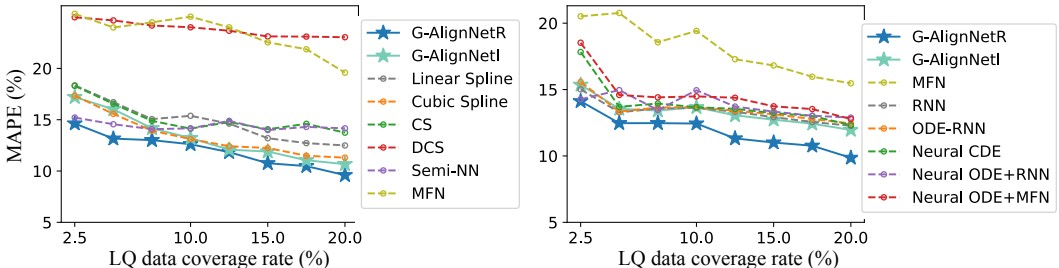

Figure 4: Results of sensitivity analysis. Left: Interpolation. Right: Extrapolation.

### 4.5 GEOMETRIC MODELING TO EXPLORE ODE STRUCTURES WITH SCARCE DATA

We present a visualization for the spiral datasets to demonstrate the capacity of approximating the underlying ODEs with limited data. Figure 5 presents the result of the three best methods. More details can be seen in Appendix D.1. The problem is challenging with scarce data (blue points) and the time-dependent magnitude for the 2-D signals. It can be found that only G-AlignNet has the ability to maintain a spiral structure and roughly learned the magnitude-increasing trend.

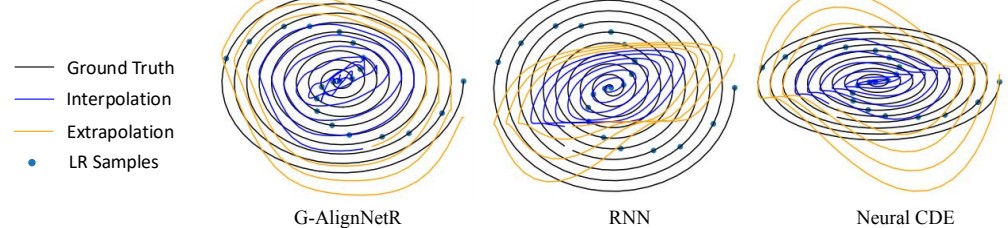

Figure 5: 2D visualization of extrapolation tasks for spiral dataset.

## 5 CONCLUSION, LIMITATION, AND FUTURE WORK

Engineering systems, due to expanding ranges, usually lack system-wide HQ data for capturing the adaptive dynamics. In this paper, we propose G-AlignNet, a unified framework designed to align LQ with HQ data while learning system dynamics. The core innovation lies in representing system dynamics through parameter flows on an orthogonal group, transforming the quality-alignment problem into a well-posed on-manifold optimization. This approach ensures global optimality and delivers superior error convergence performance. The geometric representation in G-AlignNet is fundamental, as it's linked to a broad range of on-manifold problems to meet systems demands. For instance, in the future, we will apply this representation to model system dynamics that adapt to sudden changes, which could be interpreted as a jump of a flow in Figure 1. Additionally, we will extend Optimization (2) to robust geometric optimization for highly-nonlinear systems with noisy measurements. We will also test more advanced base models like Transformers (Vaswani et al., 2017) and Mamba (Gu & Dao, 2023). A potential limitation of G-AlignNet is the longer training time compared to methods that don't use ODE solvers. However, due to the linearity of our target ODE in Equation (3), we can use a larger step size and threshold while still attaining comparable performance. For example, in our experiments, we set the relative/absolute tolerance to be $10^{-3}/10^{-4}$, whereas the default values are $10^{-5}/10^{-6}$. This configuration increases the training speed by a factor of 20. Appendix D.3 shows that our methods have relatively lower training time than other ODE-based methods.

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

## A    DATA QUALITY DEFINITION AND VISUALIZATION

In Figure 6, the four subfigures visualize the key data incompleteness issues addressed by our G-AlignNet framework. The top left subfigure illustrates high-quality data with continuous high-resolution (HR) measurements. The dataset is represented using both a line plot and scatter points, demonstrating the precision and consistency of HR data without incompleteness.

The top right subfigure presents low-resolution (LR) data, where measurements are sampled at fewer intervals due to LR sensors or downsampling to meet communication constraints (Li et al., 2024a; Willett et al., 2011). This highlights the challenges associated with sparse data acquisition. The bottom left subfigure showcases periods of data loss and communication failure caused by external events or sensor malfunction (Gill et al., 2011). These intervals are marked by gaps, visually emphasizing the absence of data over significant time periods. The bottom right subfigure illustrates random data losses, representing irregular sampling with a 30% data drop rate. Such losses often arise from sensor misconfigurations, data corruption, or human errors (Kidger et al., 2020; Chen et al., 2024; Kundu & Quevedo, 2021).

This visualization effectively categorizes and demonstrates the types of data incompleteness that G-AlignNet is designed to address, as discussed in Section 1 and 3.1.

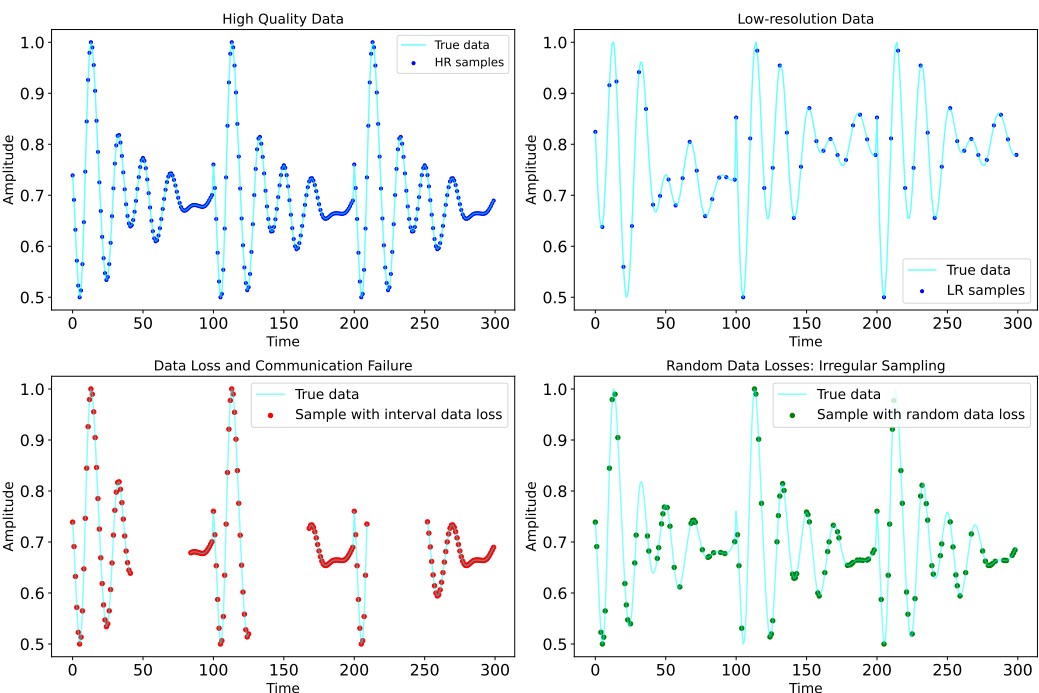

Figure 6: Visualization of different data qualities.

## B    VISUALIZATION OF DATA SIMILARITY FOR ASSUMPTION 1

**Assumption scope.** We focus on data incompleteness, the severe, common, and persistent issue in engineering and control systems. Within this study scope (i.e., data incompleteness) and assume the system has small measurement noise and low nonlinearity, Assumption 1 holds. The data property described in Assumption 1 can be affected by noise. When there are significant random factors such as sensor noise, Assumption 1 may not hold since the data similarity is reduced. In Section 3.3, we quantify the error caused by a type of noise, which demonstrates the certain robustness of our G-AlignNet. However, for more complicated noise, we need more investigations. In addition, noise can be reduced by employing more precise sensors or noise filtering techniques in engineering systems (Segovia et al., 2014).

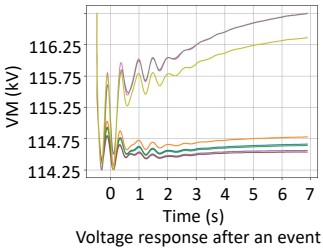 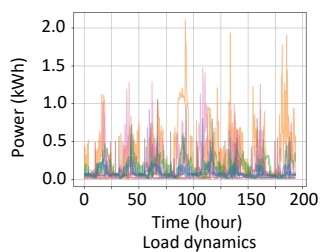 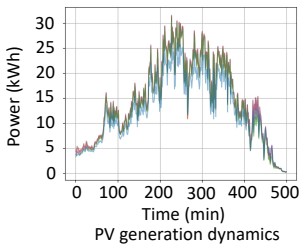

Figure 7: Data similarity for different systems.

**Assumption justification.** Our target is engineering and control systems with weakly nonlinearity and uncertainty. We assume the system is not highly nonlinear and the noise is limited. For these systems, Assumption 1 states that the high data correlations between HQ and LQ data can lead to parameter flow with the same shape but different locations on a manifold, where the shape captures similar patterns between HQ and LQ data. We have the following justifications for the validity of Assumption 1: (1) The data similarity stems from the spatial-temporal correlations and physical correlations for the system, which significantly exist in engineering systems. For example, the visualization in Fig. 7 shows that weakly nonlinear and uncertain engineering systems still have strong data correlations and similarities. (2) Under a probabilistic setting, HQ and LQ variables within a local region in the system can have high similarity due to spatial-temporal and physical correlations in both mean and variance. Then, $\Theta_x(t)$ and $\Theta_y(t)$ in our learning framework, as long as being well-trained to extract patterns of this similarity, can maintain the same shape. (3) With external forces the HQ/LQ measurements in systems usually still contain high spatial-temporal and physical correlations. For instance, when an event happens to power systems, system states (i.e., nodal voltage) have similar behaviors because of network constraints (Mai et al., 2024). The left in Fig. 7 illustrates the voltage fluctuations after an event. The PV systems or residential loads, affected by weather such as wind movements and temperature, have similar data patterns within a local region, shown in the middle and the right in Fig. 7. This is because that in a local region, the external environments are almost the same. In general, our result is very beneficial since with our methods, we only need to guarantee each local region can contain a small number of HQ sensors and can boost the quality of all LQ sensors in the region.

When the noise is limited, Assumption 1 holds for a nonlinear system because it only states the data correlations and similarity in response to disturbances between HQ and LQ data. Then, the high data correlations between HQ and LQ data can lead to parameter flow with the same shape but different locations on a manifold, where the shape captures similar patterns between HQ and LQ data. As shown in Fig. 7, weakly nonlinear and uncertain engineering systems still have strong data correlations and similarities.

**Assumption limitation** The data property described in Assumption 1 can be affected by noise. When there are significant random factors such as sensor noise, Assumption 1 may not hold since the data similarity is reduced. In Section 3.3, we quantify the error caused by a type of noise, which demonstrates the certain robustness of our G-AlignNet. However, for more complicated noise, we need more investigations. In addition, noise can be reduced by employing more precise sensors or noise filtering techniques in engineering systems (Segovia et al., 2014).

Also, when the system is highly nonlinear, Assumption 1 may cause a model incapable of representing the complicated dynamics. Under this condition, we need more investigations.

# C  DETAILED PROOFS

## C.1  PROOF OF PROPOSITION 1

**Proposition 1** (Shape Preservation via Orthogonal Matrix)**.** *If $Q \in \mathbb{R}^{n \times n}$ is an orthogonal matrix, the flow between $\Theta_x(t)$ and $\Theta_x(t)Q$ is the same.*

*Proof.* We consider two arbitrary transpose row vectors $w_i \in \mathbb{R}^n$ from $\Theta_x(t_i)$ and $w_j \in \mathbb{R}^n$ from $\Theta_x(t_j)$. By the orthogonality of $Q \in \mathbb{R}^{n \times n}$, we can show

- Length preservation: $\|Qw\|^2 = \|w\|^2$.

- Angle preservation: $(Qw_i)^\top (Qw_j) = w_i^\top (Q^\top Q) w_j = w_i^\top w_j$.

- Distance preservation: $\|Qw_i - Qw_j\| = \sqrt{(w_i - w_j)^\top (Q^\top Q)(w_i - w_j)} = \|w_i - w_j\|$.

$\square$

## C.2 PROOF OF PROPOSITION 2

**Proposition 2** (Closed-form Global Optimal Solution). *Suppose matrices $\Theta_x(t), \Theta_y(t) \in \mathcal{O}(n)$, the optimization (2) has a global minimizer $Q^* = UV^\top$, where $\frac{1}{|\mathcal{N}_y|} \sum_{i \in \mathcal{N}_y} \Theta_x(t_i)^\top \Theta_y(t_i) = U\Sigma V^\top$ is the Singular Value Decomposition.*

*Proof.* For the optimization in Equation (2), we have:

$$Q^* = \arg \min_{Q^\top Q = I} \frac{1}{|\mathcal{N}_y|} \sum_{i \in \mathcal{N}_y} \|\Theta_y(t_i) - \Theta_x(t_i)Q\|_F^2 \tag{12}$$

$$= \arg \min_{Q^\top Q = I} \frac{1}{|\mathcal{N}_y|} \sum_{i \in \mathcal{N}_y} \left[ \|\Theta_y(t_i)\|_F^2 + \|\Theta_x(t_i)Q\|_F^2 - 2\mathrm{tr}\left(Q^\top \Theta_x(t_i)^\top \Theta_y(t_i)\right) \right] \tag{13}$$

$$= \arg \min_{Q^\top Q = I} \frac{1}{|\mathcal{N}_y|} \sum_{i \in \mathcal{N}_y} -2\mathrm{tr}\left(Q^\top \Theta_x(t_i)^\top \Theta_y(t_i)\right) \tag{14}$$

$$= \arg \max_{Q^\top Q = I} \frac{1}{|\mathcal{N}_y|} \sum_{i \in \mathcal{N}_y} \mathrm{tr}\left(Q^\top \Theta_x(t_i)^\top \Theta_y(t_i)\right) \tag{15}$$

$$= \arg \max_{Q^\top Q = I} \mathrm{tr}\left(Q^\top \frac{1}{|\mathcal{N}_y|} \sum_{i \in \mathcal{N}_y} \Theta_x(t_i)^\top \Theta_y(t_i)\right). \tag{16}$$

Suppose $\frac{1}{|\mathcal{N}_y|} \sum_{i \in \mathcal{N}_y} \Theta_x(t_i)^\top \Theta_y(t_i) = U\Sigma V^\top$ is the Singular Value Decomposition, we have

$$Q^* = \arg \max_{U^\top U = I} \mathrm{tr}\left(Q^\top U\Sigma V^\top\right) \tag{17}$$

$$= \arg \max_{U^\top U = I} \mathrm{tr}\left(V^\top Q^\top U\Sigma\right) \tag{18}$$

$$= \arg \max_{Z^\top Z = I} \mathrm{tr}\left(Z\Sigma\right) \tag{19}$$

$$= \arg \max_{Z^\top Z = I} \sum_{j=1}^n Z_{jj}\sigma_j, \tag{20}$$

where we let $Z = V^\top Q^\top U$ be another orthogonal matrix. Thus, we have $\sum_{j=1}^n Z_{jj}\sigma_j \le \sum_{j=1}^n Z_{jj} \le n$. The equality holds when $Z^* = I_n$, leading to $Q^* = UV^\top$. □

## C.3  Proof of Corollary 1

**Corollary 1** (Zero-error Shape Matching). *Suppose the skew-symmetric matrix $\Omega_y(t) \equiv \Omega_x(t)$, defined in Equation (3), Optimization (2) has a unique global minimizer $Q^*$ defined in Proposition 2. The corresponding objective equals to $0$.*

*Proof.* Based on the ODE $\dot{\Theta}_x(t) = \Theta_x(t)\Omega_x(t)$ and $\dot{\Theta}_y(t) = \Theta_y(t)\Omega_y(t)$ where we denote $\Omega_x(t) \equiv \Omega_y(t) := \Omega(t)$, the parameter matrices can be represented as $\Theta_x(t) = \exp(\int_{t_0}^{t} \Omega(\tau)d\tau)\Theta_x(t_0)$ and $\Theta_y(t) = \exp(\int_{t_0}^{t} \Omega(\tau)d\tau)\Theta_y(t_0)$. Thus, we have

$$\|\Theta_x(t_i)\|_F^2 = \text{trace}\left(\Theta_x(t_i)^\top \Theta_x(t_i)\right) \tag{21}$$

$$= \text{trace}\left(\Theta_x(t_0)^\top \exp(\int_{t_0}^{t} \Omega(\tau)d\tau)^\top \exp(\int_{t_0}^{t} \Omega(\tau)d\tau)\Theta_x(t_0)\right) \tag{22}$$

$$= \|\Theta_x(t_0)\|_F^2, \forall i \tag{23}$$

since $\Omega(t)$ is a skew-symmetric matrix and thus $\exp(\int_{t_0}^{t} \Omega(\tau)d\tau) \in \mathcal{O}(n)$ is an orthogonal matrix according to Lie algebra (Lee & Lee, 2012). Similar results apply to parameters $\Theta_y(t_i)$ as $\|\Theta_y(t_i)\|_F^2 = \|\Theta_y(t_0)\|_F^2, \forall i$. Substituting the optimizer of Optimization (2) from Proposition 2, we have

$$\frac{1}{|\mathcal{N}_y|} \sum_{i \in \mathcal{N}_y} \|\Theta_y(t_i) - \Theta_x(t_i)Q^*\|_F^2 \tag{24}$$

$$= \frac{1}{|\mathcal{N}_y|} \sum_{i \in \mathcal{N}_y} \left[\|\Theta_y(t_i)\|_F^2 + \|\Theta_x(t_i)Q^*\|_F^2 - 2\text{tr}\left(Q^{*\top}\Theta_x(t_i)^\top \Theta_y(t_i)\right)\right] \tag{25}$$

$$= \frac{1}{|\mathcal{N}_y|} \sum_{i \in \mathcal{N}_y} \left[\|\Theta_y(t_0)\|_F^2 + \|\Theta_x(t_0)\|_F^2 - 2\text{tr}\left(Q^{*\top}\Theta_x(t_0)^\top \Theta_y(t_0)\right)\right] \tag{26}$$

$$= \|\Theta_y(t_0)\|_F^2 + \|\Theta_x(t_0)\|_F^2 - 2\text{tr}\left(Q^{*\top}\Theta_x(t_0)^\top \Theta_y(t_0)\right) = 0. \tag{27}$$

$\square$

## C.4 PROOF OF PROPOSITION 3

**Proposition 3** (Interpolation Error Bound). *Assume Equation (7) holds for the true parameters $\bar{\Theta}_x(t_i)$ and $\bar{\Theta}_y(t_i)$. Training G-AlignNet approximates these true parameters, thus implicitly solving the following optimization:*

$$Q^{**} = \arg \min_{Q^\top Q = I} \frac{1}{|\mathcal{N}_y|} \sum_{i \in \mathcal{N}_y} \left\| \bar{\Theta}_y(t_i) - \bar{\Theta}_x(t_i)Q \right\|_F^2. \tag{28}$$

*This leads to an estimation error of matrix $Q^{**}$, compared to true transformation matrix:*

$$\mathbb{E} \left\| Q^{**} - \bar{Q} \right\|_F \leq n^{\frac{3}{2}} \sigma_0 / \sqrt{|\mathcal{N}_y|}, \tag{29}$$

*and an interpolation error on $\tilde{\Theta}_y(t_i) = \Theta_x(t_i)Q^{**}, i \in \mathcal{N}_x \backslash \mathcal{N}_y$:*

$$\mathbb{E}_{i \in \mathcal{N}_x \backslash \mathcal{N}_y} \left\| \tilde{\Theta}_y(t_i) - \bar{\Theta}_y(t_i) \right\|_F \leq n^2 \sigma_0 / \sqrt{|\mathcal{N}_y|} + n^{\frac{1}{2}} \varepsilon_0, \tag{30}$$

*where $\varepsilon_0 = \frac{1}{|\mathcal{N}_x|} \sum_{i \in \mathcal{N}_x} \|D_i\|_F$ is the average approximation error from Neural ODE modeling.*

*Proof.* Following the conclusion in Proposition 2, we have $Q^{**} = U'V'^\top$ where the Singular Value Decomposition is given by

$$U'\Sigma'V'^\top = \frac{1}{|\mathcal{N}_y|} \sum_{i \in \mathcal{N}_y} \bar{\Theta}_x(t_i)^\top \bar{\Theta}_y(t_i) \tag{31}$$

$$= \frac{1}{|\mathcal{N}_y|} \sum_{i \in \mathcal{N}_y} \bar{\Theta}_x(t_i)^\top \bar{\Theta}_x(t_i)\bar{Q}(I_n + E_i) \tag{32}$$

$$= \frac{1}{|\mathcal{N}_y|} \sum_{i \in \mathcal{N}_y} \bar{Q}(I_n + E_i). \tag{33}$$

since $\bar{\Theta}_x(t_i) \in \mathcal{O}(n)$. Denoting $E = \frac{1}{|\mathcal{N}_y|} \sum_{i \in \mathcal{N}_y} E_i$, it satisfies that $E \sim \mathcal{N}(0, \frac{\sigma_0^2 \mathbf{1}}{|\mathcal{N}_y|})$ due to the independence of Gaussian noise. Thus, the expected estimation error of matrix $Q^{**}$ compared to the true case $\bar{Q}$ is

$$\mathbb{E} \left\| Q^{**} - \bar{Q} \right\|_F = \mathbb{E} \left\| \frac{1}{|\mathcal{N}_y|} \sum_{i \in \mathcal{N}_y} \bar{Q}(I_n + E_i) - \bar{Q} \right\|_F \tag{34}$$

$$= \mathbb{E} \left\| \bar{Q}(I + E) - \bar{Q} \right\|_F \tag{35}$$

$$= \mathbb{E} \left\| \bar{Q}E \right\|_F \tag{36}$$

$$\leq \left\| \bar{Q} \right\|_F \cdot \mathbb{E} \left\| E \right\|_F = n^{\frac{1}{2}} \cdot n\sigma_0 / \sqrt{|\mathcal{N}_y|}. \tag{37}$$

The expected interpolation error on $\tilde{\Theta}_y(t_i) = \Theta_x(t_i)Q^{**}, i \in \mathcal{N}_x \backslash \mathcal{N}_y$ is

$$\mathbb{E} \left\| \tilde{\Theta}_y(t_i) - \bar{\Theta}_y(t_i) \right\|_F = \mathbb{E} \left\| \Theta_x(t_i)Q^{**} - \bar{\Theta}_x(t_i)\bar{Q} \right\|_F \tag{38}$$

$$= \mathbb{E} \left\| \Theta_x(t_i)(\bar{Q} + \bar{Q}E) - (\Theta_x(t_i) + D_i)\bar{Q} \right\|_F \tag{39}$$

$$= \mathbb{E} \left\| \Theta_x^*(t_i)\bar{Q}E - D_i\bar{Q} \right\|_F \tag{40}$$

$$\leq \mathbb{E} \left\| \Theta_x^*(t_i)\bar{Q}E \right\|_F + \mathbb{E} \left\| D_i\bar{Q} \right\|_F \tag{41}$$

$$\leq n^{\frac{1}{2}} \cdot n^{\frac{1}{2}} \cdot n\sigma_0 / \sqrt{|\mathcal{N}_y|} + n^{\frac{1}{2}} \cdot \varepsilon_0 \tag{42}$$

$$\leq n^2 \sigma_0 / \sqrt{|\mathcal{N}_y|} + n^{\frac{1}{2}} \varepsilon_0. \tag{43}$$

$\square$

## C.5 PROOF OF PROPOSITION 4

**Proposition 4** (Data Sufficiency for Neural ODE Approximation). *The averaged approximation error in the Neural ODE model satisfies (Hillebrecht & Unger, 2022; Soetaert et al., 2012)*

$$\varepsilon_0 \leq \mathcal{O}\left(\left\|\Theta_x(t_0) - \bar{\Theta}_x(t_0)\right\|_F \cdot \frac{1}{|\mathcal{N}_x|} \cdot \frac{1 - \beta e^{\alpha T}}{1 - \beta e^{\frac{\alpha T}{|\mathcal{N}_x|}}}\right), \tag{44}$$

*where $T = t_{|\mathcal{N}_x|}$ is the end time for HQ data, $\beta > 1$ is a constant, and $\alpha$ is the largest signal value of all matrices $\Omega(t_i), i \in \mathcal{N}_x$. $h$ and $p$ are the step size and the order of the Neural ODE solver, respectively. For the adjoint method, $K$ is the number of discretized points in forward/reverse integration (Zhuang et al., 2020). $e_k^{adj} > 0$ represents the reverse inaccuracy factor in the adjoint method. $e_k^{adj} = 0$ in the naive method or Adaptive Checkpoint Adjoint (ACA) (Zhuang et al., 2020).*

Proposition 4 indicates that for a fixed time horizon $T = t_{|\mathcal{N}_x|}$, the approximation error $\varepsilon_0^*$ decreases as the volume of HQ data $|\mathcal{N}_x|$ increases.

*Proof.* According to (Hillebrecht & Unger, 2022; Soetaert et al., 2012), the approximation error between true variables $\Theta_x(t_i)$ and the learned variables $\bar{\Theta}_x(t_i)$ satisfies

$$\|D_i\|_F = \left\|\Theta_x(t_i) - \bar{\Theta}_x(t_i)\right\|_F \tag{45}$$

$$\leq \beta \left(e^{\frac{\alpha T}{|\mathcal{N}_x|}} \left\|\Theta_x(t_{i-1}) - \bar{\Theta}_x(t_{i-1})\right\|_F + \int_0^{T/|\mathcal{N}_x|} e^{\alpha(T/|\mathcal{N}_x|-s)}\delta(s)ds\right) \tag{46}$$

where $T = t_{|\mathcal{N}_x|}$ is the time horizon of HQ data. $\alpha$ is the largest signal value of all matrices $\Omega(t_i), i \in \mathcal{N}_x$, and $\beta > 1$ is a constant. $\delta(t) : \mathbb{R} \to \mathbb{R}$ is a continuous function such that $\left\|\dot{\Theta}_x(t) - \Omega(t)\bar{\Theta}_x(t)\right\|_F = \left\|\dot{D}_i\right\|_F \leq \delta(t)$. In practice, the magnitude of $\delta(t)$ is primarily determined by the truncation error of the solver used for the Neural ODE model. Since we employ the Runge-Kutta 4 (RK4) solver, a fourth-order method, the total truncation error is of order $\mathcal{O}(h^{p+1})$, where $h$ represents the step size, and $p = 4$ is the order of the ODE solver. Consequently, we can bound the magnitude of $\delta(t)$ as $|\delta(t)| \leq \mathcal{O}(h^{p+1})$ where $h \in [10^{-4}, 10^{-2}]$ in our implementation.

Notably, if we train Neural ODE with the memory-efficient adjoint method, we may come across an additional numerical error due to the mismatch of the forward/reverse hidden state trajectory (Zhuang et al., 2020). Specifically, let $K$ be the number of discretized points in the forward/reverse integration. Let $e_k^{adj} > 0$ represent the factor of numerical error introduced by the adjoint method, as shown in Equation (20) and (21) in (Zhuang et al., 2020). We derive the following error:

$$\varepsilon_0 = \underbrace{\frac{1}{|\mathcal{N}_x|}\sum_{i\in\mathcal{N}_x}\|D_i\|_F}_{\text{Average ODE global error}} + \underbrace{\mathcal{O}\left(h^{p+1}\frac{1}{|\mathcal{N}_x|}\sum_{i\in\mathcal{N}_x}\sum_{k=0}^{K-1}e_k^{adj}\right)}_{\text{Numerical error from adjoint method}} \tag{47}$$

$$\leq \frac{1}{|\mathcal{N}_x|}\sum_{i\in\mathcal{N}_x}\beta\left(e^{\frac{\alpha T}{|\mathcal{N}_x|}}\left\|\Theta_x(t_{i-1}) - \bar{\Theta}_x(t_{i-1})\right\|_F + \int_0^{T/|\mathcal{N}_x|}e^{\alpha(T/|\mathcal{N}_x|-s)}\delta(s)ds\right)$$

$$+ \mathcal{O}\left(h^{p+1}\sum_{k=0}^{K-1}e_k^{adj}\right) \tag{48}$$

$$\leq \mathcal{O}\left(\frac{1}{|\mathcal{N}_x|}\cdot\left[\sum_{i\in\mathcal{N}_x}\left(\beta e^{\frac{\alpha T}{|\mathcal{N}_x|}}\right)^{i-1}\right]\cdot\left\|\Theta_x(t_0) - \bar{\Theta}_x(t_0)\right\|_F\right) + \mathcal{O}\left(\frac{h^{p+1}}{|\mathcal{N}_x|}\cdot Te^{\alpha T/|\mathcal{N}_x|}\right)$$

$$+ \mathcal{O}\left(h^{p+1}\sum_{k=0}^{K-1}e_k^{adj}\right) \tag{49}$$

$$= \mathcal{O}\left(\frac{1}{|\mathcal{N}_x|}\cdot\left\|\Theta_x(t_0) - \bar{\Theta}_x(t_0)\right\|_F\cdot\frac{1 - \beta e^{\alpha T}}{1 - \beta e^{\frac{\alpha T}{|\mathcal{N}_x|}}} + \frac{h^{p+1}}{|\mathcal{N}_x|}\cdot Te^{\alpha T/|\mathcal{N}_x|} + h^{p+1}\sum_{k=0}^{K-1}e_k^{adj}\right). \tag{50}$$

While the last term in Equation (50) is irreducible in the adjoint method, we can utilize the naive method or more efficient methods like Adaptive Checkpoint Adjoint (ACA) (Zhuang et al., 2020) to remove this error. □

# D    MORE EXPERIMENTS

## D.1    EXPERIMENTS: SPIRAL

A spiral can be the solution of an ordinary differential equation (ODE). A common example of an ODE that has a spiral as its solution is the system of linear differential equations representing a *damped harmonic oscillator* or a *rotational system with damping*. Here's a general form of such an ODE system:

$$\begin{cases} \dfrac{dx}{dt} = ax - by \\ \dfrac{dy}{dt} = bx + ay \end{cases}, \tag{51}$$

where $a$ and $b$ are constants.

The solutions to these equations describe spirals if the real part of the eigenvalues of the corresponding coefficient matrix is negative (causing a decay towards the origin) while the imaginary part is non-zero (leading to oscillations or circular motion). For instance, for the system above, a typical solution might take the form:

$$x(t) = e^{\alpha t} \left( \cos(\omega t) + i \sin(\omega t) \right), \tag{52}$$

where $\alpha$ governs the rate of spiral decay or growth, and $\omega$ represents the frequency of oscillation. When $\alpha < 0$, the solution represents a spiral inward, and when $\alpha > 0$, it represents a spiral outward.

We define a specific format of the spiral as follows: Let $t$ represent the time variable, uniformly distributed between 0 and $20\pi$. The coordinates $x(t)$ and $y(t)$ of the spiral are generated as functions of time:

$$t \in [0, 20\pi] \tag{53}$$

The parametric equations for the spiral in polar coordinates are given by:

$$\begin{cases} x(t) = \dfrac{t \cos(t)}{20\pi} + 1 \\ y(t) = \dfrac{t \sin(t)}{20\pi} + 1 \end{cases} \tag{54}$$

Where:

- $t$ is the time variable.
- The normalization factor $20\pi$ ensures that both $x(t)$ and $y(t)$ are scaled appropriately.

We generate low-resolution time series data, with 20 points being uniformly selected from the high-resolution dataset, containing 1000 data points.

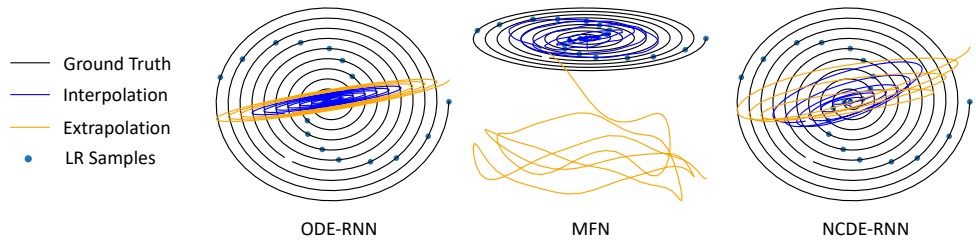

Figure 8: 2D visualization of extrapolation tasks for spiral dataset.

## D.2 ORTHOGONALITY CHECK

To validate the orthogonality of the matrix $Q$ used in our experiments, we implemented the following procedure. The function computes the deviation from orthogonality by comparing the product $Q^\top Q$ with the identity matrix.

---

**Algorithm 1** Orthogonality Check

---

**Require:** $Q_{\text{optimal}}$ (matrix to be checked)
**Ensure:** Boolean result indicating whether $Q$ is orthogonal
 1: Compute approximate identity matrix:

$$I_{\text{approx}} \leftarrow Q^\top \cdot Q$$

 2: Compute the difference between $I_{\text{approx}}$ and the true identity matrix:

$$\text{difference} \leftarrow \text{norm}(I_{\text{approx}} - I)$$

 3: Check orthogonality by verifying if $I_{\text{approx}} \approx I$:

$$\text{is\_orthogonal} \leftarrow \text{allclose}(I_{\text{approx}}, I, \text{atol} = 10^{-5})$$

   ▷ allclose is a function from the NumPy library used to compare two arrays element-wise and determine if they are equal within a specified tolerance.
 4: **if** is_orthogonal **then**
 5:     Output: "Matrix is orthogonal"
 6: **else**
 7:     Output: "Matrix is not orthogonal"
 8: **end if**

---

This procedure evaluates whether $Q$ satisfies the orthogonality condition within a specified tolerance ($10^{-6}$). The results are reported in terms of the deviation from the identity matrix and a boolean flag indicating orthogonality.

## D.3 TRAINING TIME

We have included the training costs in the following table. The results show that our methods, with a Runge-Kutta 4 (RK4) ODE solver and a relatively high tolerance, can achieve relatively moderate training time and the best model performance.

Table 3: Training time (minutes) for different systems and models.

| System | RNN | ODE-RNN | Neural CDE | MFN | NODE+RNN | NODE+MFN | G-AlignNetR | G-AlignNetI |
|---|---|---|---|---|---|---|---|---|
| Power event | 0.47 | 18.88 | 6.39 | 4.19 | 34.13 | 26.51 | 17.42 | 13.86 |
| PV | 1.10 | 53.89 | 14.85 | 9.73 | 79.31 | 62.18 | 42.71 | 30.23 |
| Load | 0.66 | 25.20 | 8.91 | 5.84 | 47.59 | 37.30 | 22.02 | 20.17 |
| Air quality | 0.70 | 27.61 | 9.46 | 6.19 | 50.47 | 36.79 | 24.04 | 18.71 |
| Spiral | 0.63 | 24.68 | 8.47 | 5.55 | 45.23 | 32.94 | 23.30 | 16.82 |

## D.4 SYSTEM DIMENSION DESCRIPTION

Our experiments were conducted on multiple systems, including the Load dataset, PV dataset, Power event dataset, Air quality dataset, and spiral dataset. The input dimension for each system is 10, 10, 6, 8, and 2. Moreover, we split HQ/LQ dimensions to be 2/8, 2/8, 1/5, 2/6, and 1/1, respectively.

## D.5 INTERPOLABLE CONTROL: STABILIZE MISSING DYNAMICS

The learned load dynamics enable us to conduct control tasks, where power generations are tuned to meet the load changes and maintain voltage stability. When there are no control actions, the teal curve shows that the fluctuations in loads can affect the voltage to exceed the safe range $[0.95, 1.05]$. When $15$min/sample data is available, the brown curve shows the best stability, close to $1.0$. G-AlignNet uses 1h data to do interpolation but obtain comparable results, shown in the blue curve.

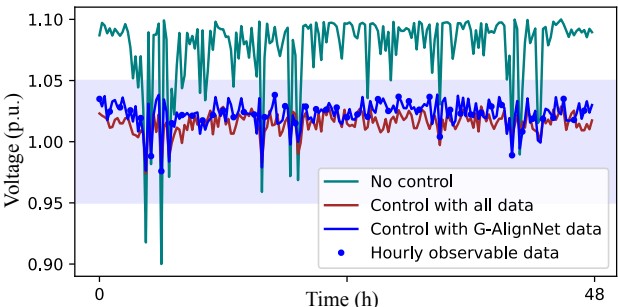

Figure 9: Stabilizing voltage using load data.

