# OpenReview forum: "G-AlignNet: Geometry-Driven Quality Alignment for Robust Dynamical Systems Modeling"
_ICLR.cc/2025/Conference — ICLR 2025 Conference Withdrawn Submission_

### Official Review · Reviewer_UQtZ · 2024-10-30

**Soundness:** 2
**Presentation:** 1
**Contribution:** 2
**Rating:** 6
**Confidence:** 3

**Summary:**

This paper presents a novel method, G-AlignNet, to align the quality between high-quality (HQ) data and low-quality (LQ) data from the geometric perspective. The method directly works on the parameter flows, using orthogonal groups as the underlying structure for the parameter manifold. This method works on data with different qualities.

**Strengths:**

- Even though I have some concerns about one proposition, the theoretical foundation of this method is still strong. The use of geometry and manifold theory provides a solid framework that offers valid guarantees for the learning process.
- The method is tested on a series of systems, demonstrating the outstanding performance of the proposed method.

**Weaknesses:**

To be honest, I am not an expert in manifold. Maybe I am wrong, but based on my understanding and research experience related to Neural ODEs, I think the following issues should be addressed or answered by authors:

- The claims regarding the limitations of Neural ODEs in LQ data are not sufficiently convincing. Specifically, the authors attribute the poor performance of Neural ODEs and their variants on LQ data to an assumption of consistent data quality. However, this explanation lacks depth and fails to convincingly argue why Neural ODEs would inherently struggle with inconsistent data.

  1. **Assumption of Consistent Data Quality**: Based on my understanding of related work and my research experience, Neural ODEs are designed to handle irregularly-sampled time series data due to their continuous nature. This characteristic inherently makes them flexible and adaptive. So I don’t think there exists the assumption of consistent data quality for Neural ODEs and their variants. The authors should provide a more detailed analysis to convince me.
  2. **References and Support**: The references provided to support these claims do not adequately substantiate the argument. The only relevant reference is [1], mentioned in the related work section, lines 80-81. However, this paper primarily focuses on addressing Neural ODEs' limitations in processing incoming data streams, rather than tackling issues related to data quality.

  In conclusion, If this is a limitation identified by the authors, a more detailed explanation and theoretical proof should be provided to justify this claim. Otherwise, The authors should either provide more suitable references that support their assertions. If these claims cannot be supported convincingly, then the motivation of this paper is really unclear.

- I have significant concerns regarding the correctness of Proposition 4, which addresses the approximation error of Neural ODEs:

  1. The proof for Proposition 4 provided in Appendix A.5 is based on [2]. However, that paper mainly discussed the approximation error of PINNs on ODEs. Since PINNs and Neural ODEs are fundamentally different [3, Section 1.1.5, Page 19], it seems inappropriate to derive the approximation error for Neural ODEs based on conclusions about PINNs, unless the authors offer more detailed theoretical justification for this connection.
  2. The approximation error of Neural ODEs is closely related to the training method used. For example, when employing the optimize-then-discretize approach (i.e., the adjoint method, as described in [4]), there is an additional discretization error that must be considered [3, Section 5.1.2.3, Page 99]. The authors do not specify which method they use for training, nor do they account for this discretization error in the proof, which could significantly affect the results.

- The experimental results appear incomplete.

  1. The authors claim that the proposed G-AlignNet can work with RNNs, INRs and PINNs. Therefore, G-AlignNet applied to these models applied to INRs should be compared with the baselines in all experiments. However, the authors only present results for one version in the tables and figures, leaving the other unreported.
  2. Regarding the extrapolation results in Figure 4, it is unclear why Neural CDE is not included as a baseline. Neural CDEs are highly relevant for extrapolation tasks and have been compared in Table 2, so it would be logical to include them in the figure for consistency and completeness.

- This paper does not clearly explain the problem formulation and experimental settings, as noted in the Questions section.

- The green points in the right Figure 3 are not explained.

**References**:

[1] Kidger, Patrick, et al. "Neural controlled differential equations for irregular time series." *Advances in Neural Information Processing Systems* 33 (2020): 6696-6707.

[2] Hillebrecht, Birgit, and Benjamin Unger. "Certified machine learning: A posteriori error estimation for physics-informed neural networks." 2022 International Joint Conference on Neural Networks (IJCNN). IEEE, 2022.

[3] Kidger, Patrick. "On neural differential equations." arXiv preprint arXiv:2202.02435 (2022).

[4] Chen, Ricky TQ, et al. "Neural ordinary differential equations." Advances in neural information processing systems 31 (2018).

**Questions:**

To be honest, I really think the writing of this paper should be improved. So many things are not explained clearly.

- The problem formulation in the paper requires more clarity, particularly regarding the definitions of HQ and LQ data.

  - I think this paper tries to train the learning model of the form $\boldsymbol{s}(t_i)=\boldsymbol{f}(\boldsymbol{s}(t_{i-1}))$, where $\boldsymbol{s}(t_i)$ is the state for the system. Let’s assume $\boldsymbol{s}(t_i) \in \mathbb{R}^d$. Based on the defintion in line 133-134, $\boldsymbol{s}=[\boldsymbol{x}, \boldsymbol{y}]$, it seems that the authors assume some variables in the state are sampled with high quality, but others are sampled with low quality, meaning that $\boldsymbol{x}\in\mathbb{R}^{d_x}$, while $\boldsymbol{y}\in\mathbb{R}^{d - d_x}$.

  - However, in lines 151-152, the authors appear to redefine $\boldsymbol{s} = [\boldsymbol{x}, \boldsymbol{y}]$ or $\boldsymbol{s} = \tilde{\boldsymbol{y}}$. So which one is correct?
  - If the correct definition is $\boldsymbol{s} = [\boldsymbol{x}, \boldsymbol{y}]$, does this imply that the authors are always assuming some variables are sampled with high quality, while others are sampled with low quality?
  - Additionally, if $\boldsymbol{y}$ is just a downsampled version of $\boldsymbol{x}$, which means $\boldsymbol{s}(t_i) = \boldsymbol{y}(t_i)$ or $\boldsymbol{s}(t_i) = \boldsymbol{x}(t_i)$ it raises the question of why we would use LQ data at all if HQ data $\boldsymbol{x}$ is already available given $\mathcal{N}_y \subset \mathcal{N}_x$.

- The authors should provide a more detailed explanation of experimental settings.

  - Although the authors have provided citations for each dataset or system used in the experiments, I believe it is still important to introduce the dimensions and size of each system. This additional information would help readers better understand the experimental settings and provide more context for evaluating the results.
  - Although the authors account for measurement noise in the theoretical analysis, it appears that noise is not addressed in the experimental section. This is particularly evident in Figure 5, where the LR samples align perfectly with the ground truth line, suggesting that noise may not have been considered in the experiments.

- This paper primarily discusses the application of G-AlignNet to RNNs and INRs. Could the proposed architecture also be applied to other common sequence models, such as Transformers or Mambas? I think such discussions can significantly improve the quality of this paper.

---

> ### Author Response · Authors · 2024-11-22
>
> $\textbf{W1: Claim of limitations for existing work}$.
>
> $\textbf{Response}$:
>
> Thank you for your question regarding the limitations of current methods and the motivation behind our approach. It is important to clarify that our work does not address all aspects of Neural ODE learning under data quality issues. Instead, we focus on a well-defined scope with specific motivations related to data quality challenges, highlighting the limitations of existing methods. To provide clarity, we emphasize the following: the study's scope, the motivation for addressing data quality issues, the limitations of existing approaches—supported by theoretical and numerical results as well as references—and the significance of our proposed method in overcoming these challenges. We add the above contents to the modified paper Introduction, Related Work, and Appendix.
>
> $\textbf{Study scope}$.
> Our paper aims to learn a dynamic model for engineering and control systems considering data quality issues, which are persistent and common in these systems. The learned dynamical model is critical for decision-making, Model Predictive Control (MPC), and model-based Reinforcement Learning (MBRL) in complex systems [20-22].
>
> $\textbf{Motivations of data quality issues}$. For many realistic systems, such as power grids, healthcare, and transportation networks [23-25], data quality fundamentally determines the extent to which dynamic information can be stored and captured in measurements for learning accurate dynamic models. For example, we focus on data incompleteness, including $(1)$ Low-Resolution (LR) measurements due to LR sensors [6] or downsampling to meet communication constraints [7]. $(2)$ A period of data losses due to communication/sensor failure, external events, etc. [8]. $(3)$ Random data losses (i.e., irregular sampling [9,10]) due to sensor configurations, data corruptions, human errors, etc. [11]. These three types are all tested in our Experiment in the first paragraph of Section 4.3. We present a clear definition in the Introduction and Section 3.1 in the revised paper. Moreover, we give visualizations for categories $(1)\sim(3)$ in Appendix A, Data Quality Definition and Visualization. Notably, data quality issues also include data inaccuracy and inconsistency with respect to the true values. However, in most physical systems, inaccurate and inconsistent measurements will be removed using mature technologies such as bad data detection [12], anomaly detection [13], noise filtering [14], etc. Consequently, these problems are converted to a data incompleteness problem. In general, data incompleteness is the central, common, and long-standing data quality issue for growing physical systems.
>
> $\textbf{Limitations of Neural ODE family}$. To learn accurate dynamic models, the family of Neural ODEs is heavily used [26-29] because of their capacity to model the continuous process using ODE solvers. This enables the process and evaluation of measurements sampled at arbitrary times. Thus, some less severe data quality issues, such as a small portion of data losses (i.e., irregularly sampled data [9,10,30]), can be properly tackled. However, this doesn't necessarily mean that all data incompleteness issues in the above categories $(1)\sim(3)$ can be fully addressed.
>
> $\textbf{Negative numerical results when learning with significant data losses}$. Significant data losses, such as low-resolution data, inherently lead to insufficient dynamic information, posing a fundamental challenge to uncovering the hidden dynamics for time intervals without samples. Under this condition, directly applying Neural ODE methods can hardly quantify the dynamic transitions within these intervals. This can lead to significant negative results for real-world systems. For example, as shown in Section 4.6, if we only know hourly load data, the voltage may not get stable between every two hourly samples, and the overly high voltage may cause overheating or damage to sensitive equipment, such as electronic devices [31]. Moreover, as shown in Table 2 in Section 4.3, for many test systems, using Neural ODE variants leads to large prediction errors ($>10\\%$ MAPE).

---

> ### Author Response · Authors · 2024-11-22
>
> $\textbf{W1: Claim of limitations for existing work}$.
>
> $\textbf{Continued response}$:
>
> $\textbf{Theoretical support}$. To substantiate our claims about the limitations of existing methods, we provide rigorous theoretical support. First, the classical theory of solving Initial Value Problems (IVPs) for ordinary differential equations (ODEs) establishes that cumulative error increases over time due to the accumulation of truncation and round-off errors, with propagation and amplification influenced by the system's dynamics and the numerical method employed [32]. High-resolution measurements mitigate these errors by enabling the IVP to be solved over smaller intervals, starting from each sample point.
>
> Second, in contrast to solving IVPs with known ODEs, Neural ODE methods introduce additional approximation dynamics during training, which can be analyzed through the framework of perturbed IVPs [33,34]. This framework shows that cumulative error persists under such settings. Our Proposition 4 also makes use of this result, which we will elaborate on in response to the next question. In summary, the derived error bounds emphasize the necessity of high-resolution data with small sampling intervals for effective error control.
>
> $\textbf{Data imputation to pre-process low-quality data}$. To address this information gap, data imputation techniques are employed to enhance data quality before using Neural ODE-based methods. These techniques leverage prior knowledge, explicit assumptions about the system's behavior, or relevant high-quality data streams to reconstruct the missing information and enhance the learning process. Model-based methods, such as multidimensional interpolation [35] and physical model-based estimations [36], rely on explicit assumptions about system behavior. Optimization-based techniques, including Compressed Sensing [37], matrix completion, and Bayesian methods [38], frame imputation as minimizing a loss function by assuming low-rank or sparsity structures. Signal processing and machine learning models offer data-driven solutions that can adapt to complex patterns [6,39], yet these often overlook domain-specific structures. Despite their utility, many existing approaches are inconsistent with the underlying data structure, as they rely on simplifying assumptions that fail to capture the intrinsic dynamics of complex systems. For example, results in Table 2, Fig. 4 and Fig. 5 in the paper show that even with the best data imputation method (i.e., Cubic Spline in our tests) as pre-processing, Neural ODE-based methods perform much worse than our G-AlignNet.
>
> Our G-AlignNet is a unified model to boost the quality and learn a good dynamic model. G-AlignNet brings significant benefits to real-world systems by using many low-quality (LQ) sensors and limited high-quality (HQ) sensors to enhance dynamic data availability for all LQ sensors and the overall dynamic model estimation.
>
> $\textbf{Why can the proposed G-AlignNet work}$? G-AlignNet effectively leverages data geometry by constructing a geometric representation that bridges high-quality (HQ) and low-quality (LQ) data. Additionally, the parameter space is carefully structured to maintain orthogonality through the use of Lie algebra, ensuring a robust foundation for geometric optimization. This geometric optimization aligns HQ and LQ data while preserving orthogonality, which enables globally optimal solutions with rigorous theoretical guarantees.
>
> Theoretical analysis shows that the above process in G-AlignNet has, to the best of the author's knowledge, the fastest convergence rate with respect to the number of LQ samples $|\mathcal{N}_y|$. Specifically, according to past analytical frameworks [15-18], errors are caused by measurement noise and the on-manifold flow approximation errors, presented as the first and the second term in the right-hand-side of Eq. (10) in the paper. Compared to the previous methods, we have the same error bound
> $\mathcal{O}(\frac{1}{\sqrt{|\mathcal{N}_y|}})$ for the error caused by noise. However, our approximation error is bounded by $\mathcal{O}(\frac{1}{|\mathcal{N}_x|})$, much smaller than the error bound in cutting-edge manifold-based compress sensing, i.e., $\mathcal{O}(\frac{1}{\log{|\mathcal{N}_y|}})$ [17]. The latter result is based on local linearization for the data manifold. Instead, G-AlignNet intelligently combines the high approximation power of ODE flows in Neural ODE and a geometric optimization with global optimality on the well-structured parameter manifold.

---

> ### Author Response · Authors · 2024-11-22
>
> $\textbf{W2: Correctness of Proposition 4}$.
>
> $\textbf{Response to point 1}$:
>
> We appreciate your careful reading for both our proposition 4 and the reference [2] in the manuscript (i.e., [33] in the response reference list).  Your misunderstanding may come from the title and final results of [2]. However, we only employ the intermediate results of [2], which are general for all Machine Learning candidate functions to approximate the underlying ODE dynamics, namely, $\hat{\phi}(\cdot)$ in Section II in [2]. To give a clear understanding, we fully explain why we utilize the intermediate theorems in [2] to prove our Proposition 4 and how the utilized theorem applies to the Neural ODE.
>
> $\textbf{Why do we employ theorems in [2] to prove the proposition 4}$?
> The proof for Proposition 4 leverages Theorem III.3 from [2], which provides an error analysis of learning parameter flows using machine-learning models as candidate functions to approximate the ODE dynamics. Importantly, we do not rely on the final theorems of [2] with additional PDE loss in PINN but instead use its intermediate results.
>
> Specifically, Theorem III.3 in [2] analyzes the prediction error $e(t) = \\|\hat{\Theta}\_{x}(t) - \Theta\_{x}(t)\\|$, where $\hat{\Theta}_x(t)$ is the machine-learned estimate of the true trajectory $\Theta_x(t)$. This theorem is broadly applicable to any machine learning method, including Neural ODEs, because it considers the general case of approximating continuous parameter flows governed by ODEs. The general error analysis comes from the classic theory of the so-called perturbed Initial Value Problem [33,34]. In this error analysis, the approximation error of the Neural ODE works as the perturbation to the IVP problem and the impact is upper bounded with moderate smoothness assumptions.
>
> In our manuscript, we model parameters $\Theta_x(t)$ with an ODE $\dot{\Theta}_x(t) = \Theta_x(t) \Omega_x(t)$, which aligns with the settings described in Equations (1) and (8) of [2]. Thus, the error bounds provided in Theorem III.3 are directly relevant to our analysis, irrespective of the specific machine learning approach used to approximate the dynamics.
>
> $\textbf{How the theorem shows the result of Neural ODE}$?
> The error bounds provided in Theorem III.3 from [2] are derived by formulating the prediction error $e(t)$ as the solution of the perturbed IVP for a perturbed ODE. Specifically, as shown in Equation (10) of [2], the evolution of the prediction error can be expressed as a perturbed IVP. By solving this IVP and applying the triangle inequality, Theorem III.3 provides an upper bound for the cumulative error.
>
> In our proof of Proposition 4, we adapt Theorem III.3 to the Neural ODE framework by directly treating the neural network in Neural ODE as the ML candidate function $\hat{\psi}(\cdot)$ in [2]. All the results still hold.

---

> ### Author Response · Authors · 2024-11-22
>
> $\textbf{W2: Correctness of Proposition 4}$.
>
> $\textbf{Continued Response to Point 2}$:
>
> We appreciate your insightful question regarding the discretization error. We agree that the discretization error, particularly when employing methods such as the optimize-then-discretize approach (e.g., the adjoint method), is critical to understanding the overall approximation error in Neural ODEs. Below, we clarify the training method used in this work and how we accounted for the discretization error in the proof of Proposition 4.
>
> $\textbf{Training Method and Discretization Error.}$
>
> In our previous proof, the first and second terms in Equation (48) in the Appendix C.5 were combined into a single $\mathcal{O}$-notation term, which might give the impression that the discretization error was omitted. However, this is not the case—we incorporated the discretization error into the derivation through $\delta(t)$.
>
> In our paper, we employed the Runge-Kutta 4 (RK4) solver for both Neural ODE training and inference. RK4 is a fixed-step explicit solver that balances computational efficiency and accuracy. As a fourth-order method, the total truncation error for RK4 is of order $\mathcal{O}(h^4)$, where $h$ is the step size.
>
> $\textbf{How We Account for Discretization Error in Proposition 4.}$ In the proof of Proposition 4, we explicitly account for the truncation error (discretization error) using the term $\delta(t):\mathbb{R}\to\mathbb{R}$ in Equation (48). This term represents a continuous function that bounds the difference between the true ODE dynamics $\dot{\Theta}_x(t)$ and the approximated dynamics $\Omega(t)\bar{\Theta}_x(t)$, as follows: $\\left\\|\\dot{\\Theta}\_x(t) - \\Omega(t)\\bar{\\Theta}\_x(t)\\right\\|\_F \le \delta(t)$.
>
> Given that the step size $h$ typically lies within the range $h \in [10^{-4}, 10^{-2}]$ in practice, and considering the Runge-Kutta 4 (RK4) solver has a truncation error of order $\mathcal{O}(h^4)$, $\delta(t)$ can be treated as a small and approximately constant upper bound. This approach is consistent with the derivation in Theorem III.3 of [2], which imposes a similar condition, $|\mathcal{R}_{\hat{\phi}}(t)| \leq \delta(t)$.
>
> $\textbf{Modifications in the revised paper to significantly improve the clarity}$. In the revised manuscript, we have implemented the following improvements to address this issue: (1) We explicitly describe the use of the RK4 solver for the Neural ODE model and emphasize the associated discretization error. (2) We have decomposed the error into two terms in Equation (49) of the revised manuscript, where the second term explicitly accounts for the discretization error. (3) Consequently, we revised Proposition 4 and its proof in the appendix to formally include the discretization error in our results. These revisions highlight that the discretization error is explicitly and rigorously addressed.
>
> $\textbf{W3: Incomplete experimental results}$.
>
> $\textbf{Response to point 1}$:
>
> In the manuscript, we use RNN as the base model because RNN is suitable for different sequential data processing, but INR focuses more on the continuous domain [40]. We appreciate the author's suggestion to add the version of G-AlignNet with INR. We use G-AlignNetR and G-AlignNetI to denote the case with RNN and INR as base models, respectively. Hence, we give additional results for G-AlignNetI. Please see the results in the modified paper. In general, G-AlignNetI works well in continuous systems like power events, air quality, and spiral datasets and achieves state-of-the-art performance with around $1\%\sim 10\%$ error reduction compared to G-AlignNetR methods. However, for systems with more uncertainty, e.g., the load and PV systems, G-AlignNetI's performance is not competitive. The main reason is that the INR model is less powerful than RNN in capturing historical trends and patterns for predictions. In the modified paper, we present all results for G-AlignNetR and G-AlignNetI in Sections 4.3 and 4.4.
>
> $\textbf{Response to point 2}$:
>
> Thank you for your careful reading and feedback. We apologize for the confusion caused by a typographical error in the legend of Fig. 4. The green dotted line represents the results of Neural CDE, not Neural ODE. Neural CDE is indeed highly relevant for extrapolation tasks, as you noted, and it was included in the comparison. Neural ODE, on the other hand, is less suitable for sequential data processing, which aligns with your observation. We will correct the legend to ensure clarity and consistency in the final version.
>
> $\textbf{W4: Problem formulation and experimental setting}$.
>
> $\textbf{Response}$:
>
>  Thank you for your insightful comment. We have provided a detailed response in the Questions section and have significantly improved the paper's clarity. We sincerely appreciate your efforts in highlighting these areas for improvement. Please refer to the revised version of the paper, and let us know if you have any further questions or suggestions.

---

> ### Author Response · Authors · 2024-11-22
>
> $\textbf{W5: Green points in Fig. 3}$.
>
> $\textbf{Response}$:
>
> Thank you for pointing this out. In Fig. 3, the green points in the left part of the Fig. represent the parameters $\Theta_y(t_i)$ corresponding to the low-resolution data points. The green points in the right panel of Fig. 3 represent the low-resolution measurements that are used in the training procedure. High-resolution measurements are not visualized as they are too dense to display effectively.
>
> The results demonstrate that G-AlignNet excels in aligning high-resolution and low-resolution measurements, extracting shared knowledge, and constructing a more accurate dynamic learning model for low-resolution data (as shown in the right panel of Fig. 3). This capability arises from G-AlignNet's ability to achieve precise parameter flow alignment (left panel of Fig. 3), which maximizes the extraction of common knowledge. The success of this alignment is attributed to our geometric representation learning approach, which leverages a well-structured parameter manifold, i.e., the orthogonal group.
>
> We hope this clarifies your concern and appreciate your attention to detail. We make all modifications accordingly in the revised paper.
>
> $\textbf{Q1: Problem formulation}$.
>
> $\textbf{Response}$:
>
> Thank you for your thoughtful comments. We acknowledge that the problem formulation in the paper lacked clarity, and we have revised the paper significantly to address these concerns. Below are point-by-point responses to your questions for clarity:
>
> $\textbf{1. Your first point is correct}$. We define the state as $\boldsymbol{s}=[\boldsymbol{x},\boldsymbol{y}]$, where measurements of the state $\boldsymbol{s}\in\mathbb{R}^{d_x+d_y}$ consist of High-Quality (HQ) measurements of $\boldsymbol{x}\in\mathbb{R}^{d_x}$ and Low-Quality (LQ) measurements $\boldsymbol{y}\in\mathbb{R}^{d_y}$. This setup is common in many real-world systems, such as power [6] and transportation [25], where economic considerations lead to HQ and LQ sensors being deployed in different parts of the system.
>
> $\textbf{2. Regarding the ambiguity in lines 151-152}$, we confirm that the correct definition is $\boldsymbol{s}=[\boldsymbol{x},\boldsymbol{y}]$. The text has been revised to consistently adhere to this definition throughout the paper. For example, we modify lines 151-152 as: "Here, $\hat{\boldsymbol{s}}(t_i)$ represents either the true measurements, which are a combination of HQ and LQ data $[\boldsymbol{x}(t_i),\boldsymbol{y}(t_i)]$ ($\forall i\in\mathcal{N}_y$), or a combination of HQ measurements and interpolated LQ data $[\boldsymbol{x}(t_i),\tilde{\boldsymbol{y}}(t_i)]$ ($\forall i\in\mathcal{N}_x\setminus\mathcal{N}_y$)."
>
> $\textbf{3. Yes, we assume that some variables are sampled with HQ and others with LQ. }$ This reflects the practical scenario where HQ measurements are only available for a subset of variables due to cost constraints, while LQ measurements are used for the remaining variables. The fixed role of HQ and LQ variables is primarily for the ease of mathematical modeling. In real-world systems, some HQ sensors may experience sensor or communication failures and temporarily function as LQ sensors. However, this dynamic does not affect our HQ-LQ alignment procedure, which is designed to make the most use of all available data regardless of the sensor status.
>
> $\textbf{4. We apologize for the confusion}$. $\boldsymbol{y}$ is not a downsampled version of $\boldsymbol{x}$.

---

> ### Author Response · Authors · 2024-11-22
>
> $\textbf{Q2: More detailed explanation for experimental setting}$.
>
> $\textbf{Response to point 1}$:
>
> Thank you for your valuable feedback. Below, we provide a more detailed explanation of the experimental setup. We add this part to Appendix D.4 in the revised paper. Our experiments were conducted on multiple systems, including the Load dataset, PV dataset, Power event dataset, Air quality dataset, and spiral dataset. The input dimension for each system is $10$, $10$, $6$, $8$, and $2$. Moreover, we split HQ/LQ dimensions to be $2/8$, $2/8$, $1/5$, $2/6$, and $1/1$, respectively.
>
>
> $\textbf{Response to point 2}$:
>
> We don't test the noise issues in our experiments because the impact of noise is quite limited in engineering systems. As noted in answer to your first question, our target data quality issue is data incompleteness, including $(1)$ Low-Resolution (LR) measurements due to LR sensors [6] or downsampling to meet communication constraints [7]. $(2)$ A period of data losses due to communication/sensor failure, external events, etc. [8]. $(3)$ Random data losses (i.e., irregular sampling [9,10]) due to sensor configurations, data corruptions, human errors, etc. [11]. These three types are all tested in our Experiment in the first paragraph of Section 4.3.  We present a clear definition the in Introduction and Section 3.1 in the revised paper. Moreover, we give visualizations for categories $(1)\sim(3)$ in Appendix A, Data Quality Definition and Visualization.
>
>
> The remaining data quality issues can be data inaccuracy and inconsistency issues, which can be caused by noises. In most physical systems, inaccurate and inconsistent measurements will be removed using mature technologies such as bad data detection [12], anomaly detection [13], noise filtering [14], etc. Consequently, these problems are converted to a data incompleteness problem. In general, data incompleteness is the central, common, and long-standing data quality issue for growing physical systems.
>
> $\textbf{Q3: Application to Transformer and Mamba}$:
>
> $\textbf{Response}$:
>
> Thank you for your insightful suggestion. Extending our on-manifold parameter flow and manifold-based geometric optimization to Transformers [41] and Mamba [42] is indeed an exciting direction, as both are powerful and widely-used models for time-series tasks. However, there are considerations regarding feasibility and computational costs. Below, we elaborate on these points:
>
> $\textbf{Feasibility of our hypernetwork structure}$. Our G-AlignNet employs a hypernetwork structure, where a Neural ODE governs the parameter flows for RNNs and INRs. This approach could, in principle, be extended to Transformers and Mamba. Notably, there are studies that explore hypernetwork-controlled Transformers, where hypernetworks facilitate task-specific adaptation in feed-forward layers [43] or value networks [44] within the attention mechanism. As for Mamba, while there is limited discussion of hypernetworks in this context, Mamba shares similarities with RNNs in its hidden state transitions. Thus, it is intuitive to extend our Neural ODE to produce the parameter flow $A(t)$ for the $A$ matrix in Mamba [42] on an orthogonal group. This can effectively capture the dynamic information because $A(t)$ governs the evolution of the hidden state dynamics.
>
> $\textbf{Computational complexity}$. While our orthogonal matrix flow framework can theoretically be integrated with any model, computational cost remains a key concern. Transformers, for instance, are computationally intensive due to the attention mechanism, and even efforts to extend attention to continuous time [10] have not fully mitigated this. Conversely, RNNs and INRs are more computationally efficient, requiring significantly less training time. Mamba, in particular, is highly efficient because of its state-space model foundation. This efficiency makes Mamba a promising candidate for integrating our geometric representation in future studies.

---

> > ### Author Response · Authors · 2024-11-22
> >
> > $\textbf{Reference}$:
> >
> > [6] Li, Haoran, et al. "Low-Dimensional ODE Embedding to Convert Low-Resolution Meters into “Virtual” PMUs." IEEE Transactions on Power Systems (2024).
> >
> > [7] Willett, Rebecca M., Roummel F. Marcia, and Jonathan M. Nichols. "Compressed sensing for practical optical imaging systems: a tutorial." Optical Engineering 50.7 (2011): 072601-072601.
> >
> > [8] Gill, Phillipa, Navendu Jain, and Nachiappan Nagappan. "Understanding network failures in data centers: measurement, analysis, and implications." Proceedings of the ACM SIGCOMM 2011 Conference. 2011.
> >
> > [9] Kidger, Patrick, et al. "Neural controlled differential equations for irregular time series." Advances in Neural Information Processing Systems 33 (2020): 6696-6707.
> >
> > [10] Chen, Yuqi, et al. "Contiformer: Continuous-time transformer for irregular time series modeling." Advances in Neural Information Processing Systems 36 (2024).
> >
> > [11] Kundu, Atreyee, and Daniel E. Quevedo. "On periodic scheduling and control for networked systems under random data loss." IEEE Transactions on Control of Network Systems 8.4 (2021): 1788-1798.
> >
> > [12] Chen, Jian, and Ali Abur. "Placement of PMUs to enable bad data detection in state estimation." IEEE Transactions on Power Systems 21.4 (2006): 1608-1615.
> >
> > [13] Ren, Hansheng, et al. "Time-series anomaly detection service at microsoft." Proceedings of the 25th ACM SIGKDD international conference on knowledge discovery & data mining. 2019.
> >
> > [14] Segovia, V. Romero, Tore Hägglund, and Karl Johan Åström. "Measurement noise filtering for PID controllers." Journal of Process Control 24.4 (2014): 299-313.
> >
> > [15] Donoho, David L., Arian Maleki, and Andrea Montanari. "The noise-sensitivity phase transition in compressed sensing." IEEE Transactions on Information Theory 57.10 (2011): 6920-6941.
> >
> > [16] Wang, Bin, et al. "Recovery error analysis of noisy measurement in compressed sensing." Circuits, Systems, and Signal Processing 36 (2017): 137-155.
> >
> > [17] Iwen, Mark A., et al. "On recovery guarantees for one-bit compressed sensing on manifolds." Discrete & computational geometry 65 (2021): 953-998.
> >
> > [18] Xu, Weiyu, and Babak Hassibi. "Compressed sensing over the Grassmann manifold: A unified analytical framework." 2008 46th Annual Allerton Conference on Communication, Control, and Computing. IEEE, 2008.
> >
> > [19] Mai, Lihao, Haoran Li, and Yang Weng. "Data Imputation with Uncertainty Using Stochastic Physics-Informed Learning." 2024 IEEE Power & Energy Society General Meeting (PESGM). IEEE, 2024.
> >
> > [20] Nagabandi, Anusha, et al. "Neural network dynamics for model-based deep reinforcement learning with model-free fine-tuning." 2018 IEEE international conference on robotics and automation (ICRA). IEEE, 2018.
> >
> > [21] Moerland, Thomas M., et al. "Model-based reinforcement learning: A survey." Foundations and Trends® in Machine Learning 16.1 (2023): 1-118.
> >
> > [22] Lenz, Ian, Ross A. Knepper, and Ashutosh Saxena. "DeepMPC: Learning deep latent features for model predictive control." Robotics: Science and Systems. Vol. 10. 2015.
> >
> > [23] Esteban, Cristóbal, Stephanie L. Hyland, and Gunnar Rätsch. "Real-valued (medical) time series generation with recurrent conditional gans." arXiv preprint arXiv:1706.02633 (2017).
> >
> > [24] Zhao, Junbo, et al. "Power system dynamic state estimation: Motivations, definitions, methodologies, and future work." IEEE Transactions on Power Systems 34.4 (2019): 3188-3198.
> >
> > [25] Zhang, Junping, et al. "Data-driven intelligent transportation systems: A survey." IEEE Transactions on Intelligent Transportation Systems 12.4 (2011): 1624-1639.
> >
> > [26] Chi, Cheng. "NODEC: Neural ODE For Optimal Control of Unknown Dynamical Systems." arXiv preprint arXiv:2401.01836 (2024).
> >
> > [27] Alvarez, Victor M. Martinez, Rareş Roşca, and Cristian G. Fălcuţescu. "Dynode: Neural ordinary differential equations for dynamics modeling in continuous control." arXiv preprint arXiv:2009.04278 (2020).
> >
> > [28] Du, Jianzhun, Joseph Futoma, and Finale Doshi-Velez. "Model-based reinforcement learning for semi-markov decision processes with neural odes." Advances in Neural Information Processing Systems 33 (2020): 19805-19816.
> >
> > [29] Shankar, Varun, et al. "Learning non-linear spatio-temporal dynamics with convolutional Neural ODEs." Third Workshop on Machine Learning and the Physical Sciences (NeurIPS 2020). 2020.
> >
> > [30] Rubanova, Yulia, Ricky TQ Chen, and David K. Duvenaud. "Latent ordinary differential equations for irregularly-sampled time series." Advances in neural information processing systems 32 (2019).

---

> > > ### Author Response · Authors · 2024-11-22
> > >
> > > $\textbf{Continued Reference}$:
> > >
> > > [31] Van Cutsem, Thierry. "Voltage instability: phenomena, countermeasures, and analysis methods." Proceedings of the IEEE 88.2 (2000): 208-227.
> > >
> > > [32] Butcher, John Charles. Numerical methods for ordinary differential equations. John Wiley & Sons, 2016.
> > >
> > > [33] Hillebrecht, Birgit, and Benjamin Unger. "Certified machine learning: A posteriori error estimation for physics-informed neural networks." 2022 International Joint Conference on Neural Networks (IJCNN). IEEE, 2022.
> > >
> > > [34] Hairer, Ernst, and Gerhard Wanner. "Solving ordinary differential equations. II, Vol. 14 of." Springer Series in Computational Mathematics (Springer Berlin Heidelberg, Berlin, Heidelberg, 1996) 10 (1996): 978-3.
> > >
> > > [35] Habermann, Christian, and Fabian Kindermann. "Multidimensional spline interpolation: Theory and applications." Computational Economics 30 (2007): 153-169.
> > >
> > > [36] Sacchi, Mauricio D., Tadeusz J. Ulrych, and Colin J. Walker. "Interpolation and extrapolation using a high-resolution discrete Fourier transform." IEEE Transactions on Signal Processing 46.1 (1998): 31-38.
> > >
> > > [37] Donoho, David L. "Compressed sensing." IEEE Transactions on information theory 52.4 (2006): 1289-1306.
> > >
> > > [38] Yi, Ming, et al. "Bayesian High-Rank Hankel Matrix Completion for Nonlinear Synchrophasor Data Recovery." IEEE Transactions on Power Systems 39.1 (2023): 2198-2208.
> > >
> > > [39] Fukami, Kai, Koji Fukagata, and Kunihiko Taira. "Machine-learning-based spatio-temporal super resolution reconstruction of turbulent flows." Journal of Fluid Mechanics 909 (2021): A9.
> > >
> > > [40] Yin, Yuan, et al. "Continuous pde dynamics forecasting with implicit neural representations." arXiv preprint arXiv:2209.14855 (2022).
> > >
> > > [41] Vaswani, A. "Attention is all you need." Advances in Neural Information Processing Systems (2017).
> > >
> > > [42] Gu, Albert, and Tri Dao. "Mamba: Linear-time sequence modeling with selective state spaces." arXiv preprint arXiv:2312.00752 (2023).
> > >
> > > [43] Mahabadi, Rabeeh Karimi, et al. "Parameter-efficient multi-task fine-tuning for transformers via shared hypernetworks." arXiv preprint arXiv:2106.04489 (2021).
> > >
> > > [44] Schug, Simon, et al. "Attention as a Hypernetwork." arXiv preprint arXiv:2406.05816 (2024).

---

> ### Comment · Reviewer_UQtZ · 2024-11-23
> **Concern about Proposition 4**
>
> Thank you for taking the time and effort to respond to my reviews. Overall, most of my questions have been addressed satisfactorily. However, I believe there is one point where the authors may have misunderstood my concern.
>
> When I referred to the additional discretization error of Neural ODEs, I was specifically discussing the error introduced in the optimize-and-discretize approach, such as the adjoint method. As noted in [1], adjoint methods can introduce gradient errors, which could impact the overall analysis.
>
> Therefore, I think it is important for the authors to clarify the following:
>
> Does GAlign-Net use adjoint methods for gradient computation?
> - If yes, this additional error should be included in the analysis.
> - If no, further clarification about the gradient computation approach is needed.
>
> Once this point is addressed, I am happy to re-evaluate my score.
>
> [1] Zhuang, Juntang, et al. "Adaptive checkpoint adjoint method for gradient estimation in Neural ODE." International Conference on Machine Learning. PMLR, 2020.

---

> > ### Author Response · Authors · 2024-11-25
> >
> > $\textbf{Response}$:
> >
> > Thank you very much for your thoughtful and thorough review of our work. We are pleased to hear that most of your concerns have been addressed to your satisfaction. We really appreciate your insightful follow-up question that points out the potential error from gradient computations. Below, we provide a detailed response to address this concern.
> >
> > $\textbf{Gradient Computation in GAlign-Net}$:
> > We confirm that in our GAlign-Net code, we don't use the adjoint method for gradient computation. Instead, we adopt the $\textbf{naive method}$ that directly back-propagates through the ODE solver, as described in [1]. Specifically, in the code, we utilize "torchdiffeq.odeint" rather than "torchdiffeq.odeint\_adjoint" for gradient computations. Additional details about the pytorch library can be found in [2]. $\textbf{This implies that our results are accurate without the additional error caused by the adjoint method}$.
> >
> > While the naive method avoids the reverse inaccuracy associated with the adjoint method, as noted in [1], it comes with higher memory consumption with a deep computation graph. Adjoint method, however, can achieve memory efficiency.
> >
> > $\textbf{Analysis of the Additional Error Caused by the Adjoint Method}$. To ensure the rigor and completeness of our proof, we have incorporated the potential gradient error introduced by the adjoint method.
> >
> > The updated analysis is presented in Section 3.3 and Appendix C.5 of the second-round revised paper. Specifically, we consider an ODE solver of order $p$ and a step size $h$. According to Equation (21) in [1], if we utilize the adjoint method, the accumulative global error has an additional numerical error due to reverse inaccuracy that corresponds to the first term on the right-hand side of Equation (21) in [1]. In our analysis of Proposition 4, we similarly compute the accumulative global error for each interval $[t_i,t_{i+1}]$, where $ i,i+1\in\mathcal{N}_x$.
> >
> > Thus, in Equation (47) of Appendix C.5 in our revision, we express the average global error $\varepsilon_0$ as the summation of two components: the average ODE global error (from numerical integration of the ODE solver) and the numerical error introduced by the adjoint method (from reverse inaccuracy), as shown in the first term ($\frac{1}{|\mathcal{N}\_x|}\sum_{i\in \mathcal{N}\_x} \left\\|D_i\right\\|_F$) and the second term ($\mathcal{O}\left(h\^{p+1} \frac{1}{|\mathcal{N}\_x|}\sum\_{i\in  \mathcal{N}\_x}\sum\_{k=0}\^{K-1} e\^{\text{adj}}\_k \right)$) in Equation (47) in Appendix C.5, respectively. Here, $e_k^{adj}$ represents the reverse inaccuracy factor in the adjoint method, equivalent to $e_k$ in Equations (20) and (21) in [1]. For naive methods or more advanced methods like Adaptive Checkpoint Adjoint (ACA) [1], $e_k^{adj}=e_k=0$.
> >
> > Through derivations, Equation (50) in Appendix C.5 identifies the term $\mathcal{O}\big(h\^{p+1}\sum\_{k=0}\^{K-1} e\^{\text{adj}}\_k\big)$ in our error bound, where $K$ is the number of discretized points in the forward/reverse integration. Equations (22) and (12) in [1] provide the explicit expression of $e\_k$ ($e\^{\text{adj}}\_k$). This term is irreducible within the error-bound framework. Consequently, we acknowledge that the adjoint method may introduce a potential error that may not converge to zero as the number of measurements increases.
> >
> > $\textbf{Overall Suggestion for G-AlignNet Implementations}$. Further research has been conducted to address the error while ensuring memory efficiency. For example, [1] provides ACA method that applies a trajectory checkpoint strategy to record the forward-mode trajectory and guarantees reverse accuracy. Therefore, we recommend incorporating such methods to enhance the training stability and memory efficiency of Neural ODEs in G-AlignNet.
> >
> > $\textbf{Reference}$:
> >
> > [1] Zhuang, Juntang, et al. "Adaptive checkpoint adjoint method for gradient estimation in neural ode." International Conference on Machine Learning. PMLR, 2020.
> >
> > [2] R. T. Q. Chen, “torchdiffeq,” 2018.

---

> > > ### Comment · Reviewer_UQtZ · 2024-11-26
> > >
> > > Thank you for your response. I am happy to raise my score to a 6. However, as I am not an expert in manifolds, the opinions of other reviewers might hold more weight.

---

### Official Review · Reviewer_Nnae · 2024-11-01

**Soundness:** 2
**Presentation:** 4
**Contribution:** 3
**Rating:** 6
**Confidence:** 3

**Summary:**

This paper introduces G-AlignNet, a model designed to handle heterogeneous dynamics data comprising both high-quality (HQ) and low-quality (LQ) measurements. Despite being sparse and noisy, LQ data shares the underlying dynamics of HQ data, allowing it to be interpolated by appropriately transforming the information from HQ data.

G-AlignNet achieves this through a time-dependent parameter flow governed by a Neural ODE for the orthogonal group. Particularly, G-AlignNet interpolates the parameter flow of LQ data by orthogonally (isometrically) transforming that of HQ dynamics. This orthogonal transformation ensures that the geometry of the parameter flow of LQ data remains invariant to that of HQ data. The constructed parameter flows for HQ and LQ data are then applied to either the RNN, INR, or another NODE to forecast the data dynamics (i.e., the main flow).

The authors present some theoretical properties of G-AlignNet, such as an error bound, and validate the model across various synthetic and real-world benchmarks.

**Strengths:**

This paper is well motivated and interesting. It is important to address the heterogeneous situations of LQ and HQ in real-world scenarios.
***
Additionally, this paper is quite novel. Although the use of parameter flow with orthogonal groups is a structure already proposed in [1], applying it to LQ data imputation is, at least to me, very interesting.
***
The authors evaluated G-AlignNet on various real-world data, demonstrating that it generally outperformed the baseline model.
***
The paper is well written and contains sufficient detail regarding the technical aspects.
***
[1] Choromanski, K. M., Davis, J. Q., Likhosherstov, V., Song, X., Slotine, J. J., Varley, J., ... & Sindhwani, V. (2020). Ode to an ODE. Advances in Neural Information Processing Systems, 33, 3338-3350.

**Weaknesses:**

I believe Assumption 1 is important, but its current form lacks sufficient support. While it is clear that the observed responses of HQ and LQ (i.e., $x(t)$ and $y(t)$) should have similar dynamical behaviors, it is less obvious why this similarity should implies that the underlying parameter flows (of deep learning models) must also align in shape. Since this is a key assumption underpinning the paper, I believe the authors should support it either theoretically or experimentally.

For example, from a theoretical perspective, could the authors derive an error bound for the LQ prediction results (i.e., the prediction of $y(t)$) by utilizing the invariance of the parameter flow?

Another suggestion is, could the authors experimentally demonstrate that the parameter flow $\Theta_y(t)$, trained with *the perfect (HQ)* $y(t)$, has an orthogonal relationship with the parameter flow $\Theta_x(t)$ for $x(t)$? This is somewhat an inverse version of the experiment of Figure 3, which shows that the shape matching encourages the better prediction results of the green LR data $y(t)$. Will training with the green *HR data* result in alignment between $\Theta_y(t)$ and $\Theta_x(t)$?

***

It is not a critical issue, but it seems that bolding is incorrect in some tables. For example, in Table 2 under the Load Data's Missing Scenario, the MAPE results show that RNN performs better, but the bolding is currently on G-AlignNet.

***

What are the definitions of $\Theta_0$ and $\Theta_1$? It seems like they might be the initial conditions of the parameter flows, but I cannot find their definitions.

**Questions:**

- Could the authors derive an error bound for the LQ prediction results (i.e., the prediction of $y(t)$) by utilizing the invariance of the parameter flow?
- Could the authors experimentally demonstrate that the parameter flow $\Theta_y(t)$, trained with *the perfect (HQ)* $y(t)$, has an orthogonal relationship with the parameter flow $\Theta_x(t)$ for $x(t)$?
- What are the definitions of $\Theta_0$ and $\Theta_1$?

---

> ### Author Response · Authors · 2024-11-22
>
> $\textbf{W1: Assumption 1 and justification}$.
>
> $\textbf{Response}$:
>
> Thank you for your thoughtful comments. We address your concerns in detail below.
>
> $\textbf{Empirical evidence of data similarity}$. We agree with you that the observed responses of HQ and LQ variables often exhibit similar dynamical behaviors due to strong spatiotemporal correlations and physical constraints. These similarities are illustrated in Appendix B through the visualization of data similarity.
>
> $\textbf{The hypothesis of parameter flow alignment}$. Based on our observations, we hypothesize that a well-designed geometric representation of the data should share the same structure in a simplified latent space, such as the parameter space governing data dynamics. In Assumption 1, the similar shapes of $\Theta_x(t)$ and $\Theta_y(t)$ represent shared knowledge extracted from HQ and LQ data. Our G-AlignNet architecture facilitates this structural alignment through restricted geometric representations and optimization, ensuring effective learning.
>
> $\textbf{Model expressiveness despite flow restrictions}$. Even with restrictions on the shape of the parameter flow, our model remains highly expressive and capable of capturing differences between HQ and LQ data. This is due to the following reasons: (1) the flow of $\Theta_x(t)$ and $\Theta_y(t)$ can reside in different regions of the manifold. (2) Eq. (1) allows for distinct static components ($\Theta_0$ and $\Theta_1$) within $\Theta$, such as bias vectors, which are designated to the dynamic learning functions of HQ and LQ data, respectively. In general, our model is highly expressive in representing different HQ and LQ data and capturing the main similarity.
>
> $\textbf{Experimental validation of parameter flow alignment}$. In Section 4.2, we demonstrate the effectiveness of our model by comparing G-AlignNet with a flow-based learning model lacking shape alignment. As shown in the right part of Fig. 3, the LQ learned dynamics (green curves) from G-AlignNet better fit the true data. This is attributed to G-AlignNet's ability to perfectly align the shapes of the parameter flows, as illustrated in the left part of Fig. 3 (note: we centralize the flows for better visualization of shape differences).
>
> $\textbf{Challenges in theoretical quantification}$. While we present extensive experimental results supporting the superiority of G-AlignNet, we acknowledge the difficulty of deriving strict error bounds for LQ predictions. This challenge arises from the nonlinearity of the base RNN or INR model, as well as the need to evaluate whether $\Theta_1$ is adequately trained. We identify this quantification as an important direction for future work and avoid making overclaims in this regard.
>
> $\textbf{W2: Validation of orthogonality}$.
>
> $\textbf{Response}$:
>
> Thanks a lot for your insightful suggestion. To verify the orthogonality between $\Theta_x(t)$ and $\Theta_y(t)$, we use the definition of matrix orthogonality and write a sub-program, shown in Section D.2 in the Appendix. Specifically, the program evaluates if a matrix $Q$ is orthogonal by computing the error $||Q^{\top}Q-I||_F$ and checking if the error is smaller than a tolerance value. Hence, we utilize the program to check each iteration of the training procedure for $Q$ in Eq. (2) in the manuscript. The results show that in each iteration, the error is around $10^{-8}\sim 10^{-7}$. Consequently, numerically we show that the orthogonal relationship is maintained.
>
> $\textbf{W3: Result bold}$.
>
> $\textbf{Response}$:
>
> We appreciate your attention to detail. We rechecked the result and find that the correct value for MAPE for RNN is $10.58 \pm 1.05$, worse than our G-AlignNet. We have carefully refined the table and verified all entries to ensure that the bolding correctly highlights the best-performing model in each scenario.
>
> $\textbf{W4: Definition of $\Theta_1$ and $\Theta_2$}$.
>
> $\textbf{Response}$:
>
> Thank you for pointing this out. $\Theta_0$ and $\Theta_1$ are distinct static components within the overall parameter set $\Theta$. These components, such as bias vectors, are specifically associated with the dynamic learning functions of HQ and LQ data, respectively. For example, for HQ and LQ RNNs, we have different bias terms in Eq. (5) in the manuscript. To clarify this further, we have now added a detailed explanation below Eq. (1) in the paper. We also proofread the complete paper to make sure all notations are properly defined. We appreciate your feedback and encourage you to refer to the updated manuscript for improved definitions.

---

> ### Author Response · Authors · 2024-11-22
>
> $\textbf{Q1: Prediction error quantification}$.
>
> $\textbf{Response}$:
>
> While we present extensive experimental results supporting the superiority of G-AlignNet, we acknowledge the difficulty of deriving strict error bounds for LQ predictions. This challenge arises from the nonlinearity of the base RNN or INR model, as well as the need to evaluate whether $\Theta_1$ is adequately trained. We identify this quantification as an important direction for future work and avoid making overclaims in this regard.
>
> $\textbf{Q2: Validation of orthogonality}$.
>
> $\textbf{Response}$:
>
> Answered in W2.
>
> $\textbf{Q3: Definition of $\Theta_0$ and $\Theta_1$}$.
>
> $\textbf{Response}$:
>
> Answered in W4.

---

> > ### Comment · Reviewer_Nnae · 2024-11-27
> >
> > Thank you for the authors' responses and the revised manuscript, particularly the additions to Appendix B and D. These provide some empirical supports for the main assumption of the paper, Assumption 1. However, I still find this validation somewhat phenomenological in nature (while I agree with the authors that theoretical proof is challenging). As reviewer SRzj pointed out, this assumption may not be sufficiently convincing for complicated non-linear systems. Based on this, I would like to maintain my current score.

---

### Official Review · Reviewer_SRzj · 2024-11-03

**Soundness:** 2
**Presentation:** 2
**Contribution:** 2
**Rating:** 5
**Confidence:** 4

**Summary:**

This manuscript presents a method to handle data of inconsistent quality for the modeling of dynamic systems. The method draws influence from geometric method of the parameter manifold, in particular weight matrix flow-based geometric representation. Extensive experimental results are presented in the manuscript.

**Strengths:**

The manuscript tries to address an important problem. The author(s) cited a few practical applications, e.g., the residential electricity consumption and photovoltaic systems, as well as power system event measurement dataset. The paper is written reasonably well. The theoretical contribution appears to be solid.

**Weaknesses:**

My biggest concern is the motivation and problem definition of the paper. The manuscript appears to address low-quality and high-quality of data, but the author(s) never provided a clear definition how the "quality" of the data is defined. In line 130, it is only indicated that the amount of low quality data can be much larger than the number of high quality data. Near line 273, two potential causes of *low quality* are presented: 1) sensor noise, and 2) the *approximate error* of the Neural ODE. Near line 364, the term *Low Resolution* and *High Resolution* are used. One obvious question is what the relationship between LQ/HQ and LR/HR is. Finally near line 385, it appears that the so-called *LR* data are generated by dropping certain number of data points. The problem definition gradually downgrades from *Quality of data* to *Rate of data*, to *dropping some data*. Although even with dropping data, it could be a very interesting problem in practice, and the topic is well deserve a treatise, it is not the paper as presented here.

Related to the *quality* of data, the Assumption 1 in line 156 appears very strong. If the problem definition is indeed the data with different and unequal sampling rate, intuitively Assumption 1 holds. If the *quality* of the data is represented not only by sampling rate, but also with sensor noise, Assumption 1 is too strong and can very well be invalid. One counter example I can give is that unless the system is inherently linear, and there is no external forcing terms, then the additive noise to the sensor data may well cause different trajectory.  The consideration of data vs physics for modeling dynamical systems have been discussed in prior publications, especially from the perspective of uncertainties (both aleatoric and epistemic)

**Questions:**

Reiterate my concerns related to the definition of *data quality*, please consider:

1. Provide a precise definition of "data quality" early in the paper
2. Clarify if and how concepts like resolution, sampling rate, and noise relate to their definition of quality
3. Consistently use terminology throughout the paper when referring to data quality
4. Explain the relationship between the theoretical framework and the experimental setup, particularly how "dropping data points" relates to their notion of data quality

Related to the validity of Assumption 1, please consider:

1. Clarify what aspects of data quality (e.g. sampling rate, noise) are covered by Assumption 1
2. Discuss the limitations of this assumption, particularly for nonlinear systems or in the presence of sensor noise
3. Provide justification for why this assumption is reasonable for their target applications, or acknowledge where it may break down
4. Consider addressing the specific questions about how Assumption 1 holds for nonlinear systems with sensor noise and/or linear systems with external forcing terms

---

> ### Author Response · Authors · 2024-11-22
>
> $\textbf{W1: Data quality definition}$.
>
> $\textbf{Response}$:
>
> We really appreciate your concern about the scope of the problem we can tackle. Hence, we follow your suggestion to give a clear definition of our target data-quality issues. Then, we emphasize why our G-AlignNet is capable of tackling these problems and is the state-of-the-art method. Finally, we make clarifications in the revised paper for your detailed comments.
>
> $\textbf{Definitions for data quality issues}$. In general, our G-AlignNet aims to tackle the most severe and persistent issue, $\textbf{data incompleteness}$, in control and engineering systems. Data incompleteness refers to the absence of values in the dataset. More specifically, the incompleteness can be categorized into $(1)$ Low-Resolution (LR) measurements due to LR sensors [6] or downsampling to meet communication constraints [7]. $(2)$ A period of data losses due to communication/sensor failure, external events, etc. [8]. $(3)$ Random data losses (i.e., irregular sampling [9,10]) due to sensor configurations, data corruptions, human errors, etc. [11]. These three types are $\textbf{all tested in our Experiment}$ in the first paragraph of Section 4.3. We present a clear definition in the Introduction and Section 3.1 in the revised paper. Moreover, we $\textbf{give visualizations for categories $(1)\sim(3)$}$ in Appendix A, Data Quality Definition and Visualization.
>
> Notably, data quality issues also include $\textbf{data inaccuracy and inconsistency}$ with respect to the true values. However, in most physical systems, inaccurate and inconsistent measurements will be removed using mature technologies such as bad data detection [12], anomaly detection [13], noise filtering [14], etc. Consequently, these problems are converted to a data incompleteness problem. In general, data incompleteness is the central, common, and long-standing data quality issue for growing physical systems.
>
> $\textbf{Why is G-AlignNet state-of-the-art to tackle all data incompleteness problems in categories $(1)\sim(3)$}$? G-AlignNet can tackle all data incompleteness problems using a unified framework. This is because training G-AlignNet essentially learns the geometric matrix flows $\Theta_x(t)$ and $\Theta_y(t)$, which are further utilized to solve an optimal quality-alignment problem in Eq. (2) in the manuscript. The optimization demands the evaluation of the matrix flow at Low-Quality (LQ) observable times $\\{t_i\\}\_{i\\in\\mathcal{N}\_y}$. Luckily, the matrix flow is continuous and can be evaluated at arbitrary times with the help of ODE solvers, no matter how $\\{t_i\\}_{i\in\mathcal{N}_y}$ behaves (such as irregular interval, random drop, etc.) in categories $(1)\sim(3)$.
>
> Thus, all parameter matrices evaluated at $\\{t_i\\}_{i\in\mathcal{N}_y}$ can be inputted to the optimization. Then, the optimization outputs optimal transformation matrix $Q^{\ast}$ that converts $\Theta_x(t_i)$ at HQ observable time $\\{t_i\\}\_{i\in\mathcal{N}\_x}$ to generate high-quality values for $\Theta_y(t)$. Namely, the output is $\tilde{\Theta}\_y(t\_i)$ at HQ time $\\{t_i\\}\_{i\in\mathcal{N}\_x}$ that can be guaranteed to have good approximations with a small error bound.
>
> Moreover, theoretical analysis shows that the above process in G-AlignNet has, to the best of the author's knowledge, the fastest convergence rate with respect to the number of LQ samples $|\mathcal{N}_y|$. Specifically, according to past analytical frameworks [15-18], errors are caused by measurement noise and the on-manifold flow approximation errors, presented as the first and the second term in the right-hand-side of Eq. (10) in the paper. Compared to the previous methods, we have the same error-bound
> $\mathcal{O}(\frac{1}{\sqrt{|\mathcal{N}_y|}})$ for the error caused by noise. However, our approximation error is bounded by $\mathcal{O}(\frac{1}{|\mathcal{N}_x|})$, much smaller than the error bound in cutting-edge manifold-based compress sensing, i.e., $\mathcal{O}(\frac{1}{\log{|\mathcal{N}_y|}})$ [17]. The latter result is based on local linearization for the data manifold. Instead, G-AlignNet intelligently combines the high approximation power of ODE flows in Neural ODE and a geometric optimization with global optimality on the well-structured parameter manifold (i.e., orthogonal group). In particular, the structure is, by design, embedded into our proposed representation learning using Lie algebra.

---

> > ### Comment · Reviewer_SRzj · 2024-11-26
> >
> > thanks for the detailed responses and revision.
> >
> > The revision made it clear that the term *quality* as in *high quality* and *low quality* reflects the amount of missing data and/or sampling rate.  Under this definition, I can see assumption 1 can be justified.  However, I am still not convinced that Assumption 1 holds when the system is strongly nonlinear and the sensor noise is considered.  The method is only true for *stable* and *linear* (or *weakly nonlinear*) systems, as demonstrated in Appendix D. Whether the proposed method works for general nonlinear systems with strong sensor noise needs to be carefully investigated. It is also my belief that even in the narrowed definition, the proposed method is still valuable.
> >
> > In my opinion, the paper should be *significantly* revised to narrow the scope. To focus on dynamic system modeling as:
> > * modeling dynamical systems in the context of Neural ODE with missing data or lower sampling rate
> > * with no sensor noise or tightly controlled SNR in sensor noise
> >
> > I maintain my score and encourage the author(s) to consider the proposed revision.

---

> > > ### Author Response · Authors · 2024-11-27
> > >
> > > $\textbf{Response:}$
> > >
> > > We sincerely thank the reviewer for their insightful feedback and for highlighting the importance of clarifying the scope and limitations of our work. Based on the reviewer's suggestion, we narrowed the study scope and significantly revised the paper. Please see the submitted third-round revision. Then, we make the following clarifications.
> > >
> > > $\textbf{Acknowledging the Limitations:}$ We appreciate the reviewer’s observation regarding the limitations of our method in addressing strongly nonlinear systems and scenarios with significant measurement noise. We agree with this limitation. For example, in our previous revision, we explicitly noted the need for limited noise in the response, the description following Assumption 1 in Section 3.1, and the final paragraph of Appendix B.
> > >
> > > For the nonlinearity, we note that the data similarity and the same shape of parameter flow ($\Theta_x(t)$ and $\Theta_y(t)$) in Assumption 1 are valid for weakly nonlinear and stable systems, as demonstrated in our experiments. This is because our learning model is known to be nonlinear, capable of processing simple $\Theta_x(t)$ and $\Theta_y(t)$ with the same shape and reconstructing weakly nonlinear dynamics. However, when the system is highly nonlinear, Assumption 1 may cause a model incapable of representing the complicated dynamics. Under this condition, we need more investigations.
> > >
> > > In this revision, we have made substantial updates to further acknowledge this limitation, and clarify and refine the scope of our study, ensuring that the assumptions and limitations are more transparently communicated.
> > >
> > > $\textbf{Addressing the Limitation in Future Work}$. Since our model has well-structured geometry and optimization, it is promising to extend our approach to robustly address measurement noise and high nonlinearities. For instance, the geometric optimization formulation (Optimization (2)) in our paper could potentially be adapted to incorporate robust optimization techniques that mitigate the effects of noise.
> > >
> > > $\textbf{Focusing Our Strength to Solve A Valuable Problem}$. We agree with the reviewer’s suggestion that we should focus on the strengths of our method—such as addressing missing or low-resolution data, which is a valuable and meaningful contribution to the field. By learning a well-structured geometric representation, our G-AlignNet method generates globally optimal solutions for multi-resolution quality alignment while maintaining high model expressivity. The model is innovative and brings excellent theoretical and numerical results for data imputation and dynamic model learning.

---

> > > > ### Author Response · Authors · 2024-11-27
> > > >
> > > > $\textbf{Continued Response}$:
> > > >
> > > > $\textbf{Significantly Refine the Paper}$. To address the reviewer's concern, we have revised the second-round revision to explicitly clarify and narrow the scope of our work. The primary focus of this work is addressing data incompleteness in dynamic system modeling, particularly scenarios involving missing data or low-resolution sampling. We do not claim applicability to highly nonlinear systems or scenarios with significant measurement noise. By refining the scope, we emphasize the practical use cases where our method is most effective and impactful.
> > > >
> > > > Specifically, we give clarification in the following places.
> > > >
> > > > (1) In the second paragraph of the Introduction, we emphasize that we tackle data incompleteness. Also, we add the following statement: "We restrict our analysis to systems with low nonlinearity and limited noise, prioritizing the challenge of handling significant missing data."
> > > >
> > > > (2) We modify Assumption 1 to emphasize the low nonlinearity and small noise. Specifically, we write: "Assume a system with low nonlinearity and limited measurement noise. The HQ and LQ states of the system, $\boldsymbol{x}(t)$ and $\boldsymbol{y}(t)$, exhibit high similarity. Therefore, the flows of $\Theta_x(t)$ and $\Theta_y(t)$ share the same shape but occupy different locations on the manifold $\mathcal{M}$."
> > > >
> > > > (3) In Section 3.3 Theoretical Analysis, we improve the explanations after Proposition 3. Specifically, we add the following illustrations. "The error bound from noise indicates that our model is robust to Gaussian noise with low variance. We need further investigations into the model's performance under high noise levels."
> > > >
> > > > (4) In the Experiment Setting, we add the following illustrations. "Our test systems have weakly nonliearity and no measurement noise. However, the available data amount largely varies to create data incompleteness."
> > > >
> > > > (5) In Conclusion and Future Work, we highlight that future work will focus on extending the study to nonlinear systems with significant sensor noise. Specifically, we add "Additionally, we will extend Optimization (2) to robust geometric optimization for highly-nonlinear systems with noisy measurements."
> > > >
> > > > (6) In Appendix B, we emphasize that we consider weakly nonlinear system with limited measurement noise in $\textbf{Assumption scope}$ and $\textbf{Assumption justification}$. In $\textbf{Assumption limitation}$, we add the following paragraph. "Also, when the system is highly nonlinear, Assumption 1 may cause a model incapable of representing the complicated dynamics. Under this condition, we need more investigations."
> > > >
> > > > We sincerely appreciate the reviewer’s valuable suggestions and have made substantial revisions to the paper to clarify its scope. Please let us know if you have any additional questions or feedback. We look forward to engaging in further discussion.

---

> > > > > ### Author Response · Authors · 2024-11-30
> > > > > **Follow-Up on Rebuttal Response and Gratitude for Your Feedback**
> > > > >
> > > > > Dear Reviewer SRzj:
> > > > >
> > > > > I hope this message finds you well. I wanted to take a moment to express my gratitude for your thoughtful feedback on our paper and the time you’ve dedicated to the review process. In our latest rebuttal, we have carefully addressed your concerns regarding the definition of data quality and the scope of validity for Assumption 1. Your insights have been invaluable in enhancing the clarity and rigor of our work, and we truly appreciate them.
> > > > >
> > > > > I understand the past two days were the Thanksgiving holidays, and I hope you had a relaxing and enjoyable break. If you could kindly take a moment to review our response and share your thoughts, we would greatly appreciate it. Please don’t hesitate to let us know if there’s any additional clarification or further information we can provide.

---

> > > > > > ### Comment · Reviewer_SRzj · 2024-12-02
> > > > > >
> > > > > > Thanks very much for the effort to revise the manuscript. After comparing the 3rd and 2nd revision, my view is that more substantial revision is needed, especially to narrow the scope of the paper.  The author(s) should consider revising the general description of the proposed method, also rethink more representative experiments besides Table 1 and 2, and further expand the experiments in Fig 4. Nevertheless, I believe the work is sound. I revise my score slightly.

---

> ### Author Response · Authors · 2024-11-22
>
> $\textbf{W1: Data quality definition}$.
>
> $\textbf{Continued response}$:
>
> $\textbf{Clarifications of your proposed details}$. First, we modify line 130 by specifying our target data incompleteness with a clear definition and three categories. The definition and visualization are also presented in Appendix A to make readers easily understand the target problems. Then, we show that all issues indicate the differences in data amount. Mathematically,  $\mathcal{N}_y\subset \mathcal{N}_x$ and $|\mathcal{N}_y|\ll |\mathcal{N}_x|$. Second, in line 273, the measurement noise and approximation error are used as two sources to analyze the final error bound. In addition to data incompleteness, the analytical result suggests that our method is robust to linear measurement noise, similar to Compressed Sensing. Third, in line 364, we use Low-Resolution (LR) and High-Resolution (HR) (i.e., category $(1)$ in the previous statement) as an example to visualize parameter flow, which reveals that our G-AlignNet can achieve perfect shape match. The general results for categories $(1)\sim (3)$ are presented in Section 4.3. Finally, in line 385, we present three categories, which are achieved by dropping data for low resolutions, dropping consecutive intervals, and dropping data randomly. Our results in Tables 1 and  2 show that in most datasets for the three quality issues, G-AlignNet has the best performance.
>
> $\textbf{W2: Clarification of Assumption 1}$.
>
> $\textbf{Response}$:
>
> Thanks for your comments. We provide the following thorough explanations for the validity, limitations, and justifications of Assumption 1. Related contents are included in the description below Assumption 1 and Appendix B in the modified paper.
>
> $\textbf{Assumption 1 validity under data incompleteness and noise impact}$. First, as we explained in the previous answer, we focus on data incompleteness, the severe, common, and persistent issue in engineering and control systems. We agree with you that within this study scope (i.e., data incompleteness), Assumption 1 holds.
>
> $\textbf{Limitations of Assumption 1}$. Second, we admit that the data property described in Assumption 1 can be affected by noise. When there are significant random factors such as sensor noise, Assumption 1 may not hold since the data similarity is reduced. In Section 3.3, we quantify the error caused by a type of noise, which demonstrates the certain robustness of our G-AlignNet. However, for more complicated noise, we need more investigations. In addition, noise can be reduced by employing more precise sensors or noise filtering techniques in engineering systems [14].
>
> $\textbf{Justifications for Assumption 1 under high data uncertainty}$. Third, we give some justifications for the validity of Assumption 1 under high data uncertainty: (1) $\Theta_x(t)$ and $\Theta_y(t)$ in Assumption 1 can be naturally extended to a probabilistic setting, as pointed out in the first paragraph of Section 3.1. These parameters can represent neural network weights to approximate both the mean and the variance. Hence, G-AlignNet has the capacity to capture probabilistic dynamics. (2) Under a probabilistic setting, HQ and LQ variables within a local region in the system can have high similarity due to spatial-temporal and physical correlations in both mean and variance. Then, $\Theta_x(t)$ and $\Theta_y(t)$ in our learning framework, as long as being well-trained to extract patterns of this similarity, can maintain the same shape. In general, when the noise is limited, Assumption 1 holds for a nonlinear system because it only states the data correlations and similarity in response to disturbances between HQ and LQ data. Then, the high data correlations between HQ and LQ data can lead to parameter flow with the same shape but different locations on a manifold, where the shape captures similar patterns between HQ and LQ data.  As shown in Appendix B, Visualization of Data Similarity, highly nonlinear and uncertain engineering systems still have strong data correlations and similarities.
>
> $\textbf{Q1: Definition of data quality}$.
>
> $\textbf{Response}$:
>
> In general, our G-AlignNet mainly tackles $\textbf{data incompleteness}$, in control and engineering systems. Data incompleteness refers to the absence of values in the dataset, including $(1)$ Low-Resolution (LR) measurements, $(2)$ a period of data losses, and $(3)$ random data losses (i.e., irregular sampling [9,10]). Mathematically,  $\mathcal{N}_y\subset \mathcal{N}_x$ and $|\mathcal{N}_y|\ll |\mathcal{N}_x|$. In the revised paper, we add the definition to the Introduction and mathematical explanations to Section 3.1. Finally, we give visualizations for categories $(1)\sim(3)$ in Appendix A, Data Quality Definition and Visualization. The definition is added to the Introduction in the revised paper.

---

> ### Author Response · Authors · 2024-11-22
>
> $\textbf{Q2: Clarification to resolution, sampling rate, etc.}$.
>
> $\textbf{Response}$:
>
> As explained in previous questions, we tackle $\textbf{data incompleteness}$, including LR data, a period of data losses due to communication failure, and random data losses. Data noise belongs to the $\textbf{data inaccuracy}$ issue, and there is another $\textbf{data inconsistency}$ issue. In most physical systems, inaccurate and inconsistent measurements will be removed using mature technologies such as bad data detection [12], anomaly detection [13], noise filtering [14], etc. Consequently, these problems are converted to a data incompleteness problem. In general, data incompleteness is the central, common, and long-standing data quality issue for growing physical systems. We clarify these statements in the Introduction of the revised paper.
>
> $\textbf{Q3: Consistent terminology }$.
>
> $\textbf{Response}$:
>
> In the revised paper, we utilize High/Low-Quality (HQ/LQ) data to denote overall quality differences and emphasize that we focus on data completeness quality. In Experiment, we utilize High/Low-Resolution (HR/LR) data to denote the quality difference associated with the different sampling resolutions (category (1) in previous answers). We utilize missing intervals to denote the low quality associated with the absence of the interval data (category (2) in previous answers). We utilize irregularly sampled data (or data with random drops) to denote the low quality associated with random data losses. We note that these two terms are frequently and almost equivalently used in  [9,10]. We clarify these terminologies throughout the revised paper.
>
> $\textbf{Q4: Relation to experimental setup}$.
>
> $\textbf{Response}$:
>
> In our Experiment, the number of dropped data points is equal to $|\mathcal{N}_x|-|\mathcal{N}_y|$. Hence, given fixed HQ data, if the data drop rate is higher, $|\mathcal{N}_x|-|\mathcal{N}_y|$ is higher, the LQ data amount $|\mathcal{N}_y|$ is lower, the data quality is worse, and the obtained MSE/MAPE error should be generally larger. In Section 3.3, we quantify that our method's error bound is around $\mathcal{O}(\frac{1}{\sqrt{|\mathcal{N}_y|}} + \frac{1}{|\mathcal{N}_x|})$, which is the lowest among other data interpolation methods. In Fig. 4, we plot the error with respect to the LQ data coverage rate ($\frac{|\mathcal{N}_y|}{|\mathcal{N}_x|}$, proportional to $|\mathcal{N}_y|$ given fixed $|\mathcal{N}_x|$ ), which approximately aligns with the theoretical results. We add the above explanations to the Experiment in the revised paper.
>
> $\textbf{Q5: Aspect of data quality to Assumption 1}$.
>
> $\textbf{Response}$:
>
> As we explained in the previous answer, we focus on data incompleteness, the severe, common, and persistent issue, in engineering and control systems. We agree with you that within our study scope (i.e., data incompleteness), Assumption 1 holds. For systems with high uncertainty, we can still give some justifications for the validity of Assumption 1: (1) $\Theta_x(t)$ and $\Theta_y(t)$ in Assumption 1 can be naturally extended to a probabilistic setting, as pointed out in the first paragraph of Section 3.1. These parameters can represent neural network weights to approximate both the mean and the variance. Hence, G-AlignNet has the capacity to capture probabilistic dynamics. (2) Under a probabilistic setting, HQ and LQ variables within a local region in the system can have high similarity due to spatial-temporal and physical correlations in both mean and variance. Then, $\Theta_x(t)$ and $\Theta_y(t)$ in our learning framework, as long as being well-trained to extract patterns of this similarity, can maintain the same shape. We treat the investigation of the probabilistic setting in future work. Finally, to make readers easily understand Assumption 1, we visualize the realistic datasets to demonstrate the data similarity. The visualization is introduced in Appendix B, Visualization of Data Similarity in the revised paper. We add the above clarification to Appendix B in the revised paper.

---

> ### Author Response · Authors · 2024-11-22
>
> $\textbf{Q6: Limitations of Assumption 1}$.
>
> $\textbf{Response}$:
>
> $\textbf{Impact of Noise on Assumption 1.}$
> We admit that the data property described in Assumption 1 can be affected by noise. When there are significant random factors such as sensor noise, Assumption 1 may not hold since the data similarity is reduced. In Section 3.3, we quantify the error caused by a type of noise, which demonstrates the certain robustness of our G-AlignNet. However, for more complicated noise, we need more investigations. In addition, noise can be reduced by employing more precise sensors or noise filtering techniques in engineering systems [14]. We add the limitation to Appendix B of the revised paper.
>
> $\textbf{Assumption 1 holds when noise is limited.}$ When the noise is limited, Assumption 1 holds for a nonlinear system because it only states the data correlations and similarity in response to disturbances between HQ and LQ data. Then, the high data correlations between HQ and LQ data can lead to parameter flow with the same shape but different locations on a manifold, where the shape captures similar patterns between HQ and LQ data. As shown in Appendix B, Visualization of Data Similarity, highly nonlinear and uncertain engineering systems still have strong data correlations and similarities.  We add the above clarification to the description of Assumption 1 in the revised paper.
>
> $\textbf{Q7: Justification for Assumption 1}$.
>
> $\textbf{Response}$:
>
> $\textbf{Assumption 1 holds in many nonlinear and uncertain engineering systems.}$
> Our target is engineering and control systems with nonlinearity and uncertainty. For these systems, Assumption 1 states that the high data correlations between HQ and LQ data can lead to parameter flow with the same shape but different locations on a manifold, where the shape captures similar patterns between HQ and LQ data.
>
> We have the following justifications for the validity of Assumption 1: (1) The data similarity stems from the spatial-temporal correlations and physical correlations for the system, which significantly exist in engineering systems. For examples, the visualization in Appendix B shows that highly nonlinear and uncertain engineering systems still have strong data correlations and similarity. (2) Under a probabilistic setting, HQ and LQ variables within a local region in the system can have high similarity due to spatial-temporal and physical correlations in both mean and variance. Then, $\Theta_x(t)$ and $\Theta_y(t)$ in our learning framework, as long as being well-trained to extract patterns of this similarity, can maintain the same shape. (3) With external forces the HQ/LQ measurements in systems usually still contain high spatial-temporal and physical correlations. For instance, when an event happens to power systems, system states (i.e., nodal voltage) have similar behaviors because of network constraints [19]. The left Figure in Appendix B illustrates the voltage fluctuations after an event. The PV systems or residential loads, affected by weather such as wind movements and temperature, have similar data patterns within a local region, shown in the middle and the right Figure in Appendix B.
>
> $\textbf{Scenarios that Assumption 1 might break down.}$
> When there are significant random factors such as sensor noise, Assumption 1 may not hold since the data similarity is reduced. In Section 3.3, we quantify the error caused by a type of noise, which demonstrates the certain robustness of our G-AlignNet. However, for more complicated noise, we need more investigations. In addition, noise can be reduced by employing more precise sensors or noise-filtering techniques in engineering systems [14]. We add the justification to Appendix B of the revised paper.
>
> $\textbf{Q8: Assumption for external noise}$.
>
> $\textbf{Response}$:
>
> Assumption 1 relies on data similarity. As explained in previous answers, for nonlinear systems with significant random sensor noise, data similarity in Assumption 1 may not hold. When engineering systems have external forcing terms, the HQ/LQ measurements in systems usually still contain high spatial-temporal and physical correlations. For instance, when an event happens to power systems, system states (i.e., nodal voltage) have similar behaviors because of network constraints [19]. The left Figure in Appendix B illustrates the voltage fluctuations after an event. The PV systems or residential loads, affected by weather such as wind movements and temperature, have similar data patterns within a local region, shown in the middle and the right Figure in Appendix B. This is because, in a local region, the external environments are almost the same. In general, our result is very beneficial since, with our methods, we only need to guarantee each local region can contain a small number of HQ sensors and can boost the quality of all LQ sensors in the region. We add the justification to Appendix B of the revised paper.

---

> > ### Author Response · Authors · 2024-11-22
> >
> > $\textbf{References}$.
> >
> > [6] Li, Haoran, et al. "Low-Dimensional ODE Embedding to Convert Low-Resolution Meters into “Virtual” PMUs." IEEE Transactions on Power Systems (2024).
> >
> > [7] Willett, Rebecca M., Roummel F. Marcia, and Jonathan M. Nichols. "Compressed sensing for practical optical imaging systems: a tutorial." Optical Engineering 50.7 (2011): 072601-072601.
> >
> > [8] Gill, Phillipa, Navendu Jain, and Nachiappan Nagappan. "Understanding network failures in data centers: measurement, analysis, and implications." Proceedings of the ACM SIGCOMM 2011 Conference. 2011.
> >
> > [9] Kidger, Patrick, et al. "Neural controlled differential equations for irregular time series." Advances in Neural Information Processing Systems 33 (2020): 6696-6707.
> >
> > [10] Chen, Yuqi, et al. "Contiformer: Continuous-time transformer for irregular time series modeling." Advances in Neural Information Processing Systems 36 (2024).
> >
> > [11] Kundu, Atreyee, and Daniel E. Quevedo. "On periodic scheduling and control for networked systems under random data loss." IEEE Transactions on Control of Network Systems 8.4 (2021): 1788-1798.
> >
> > [12] Chen, Jian, and Ali Abur. "Placement of PMUs to enable bad data detection in state estimation." IEEE Transactions on Power Systems 21.4 (2006): 1608-1615.
> >
> > [13] Ren, Hansheng, et al. "Time-series anomaly detection service at microsoft." Proceedings of the 25th ACM SIGKDD international conference on knowledge discovery & data mining. 2019.
> >
> > [14] Segovia, V. Romero, Tore Hägglund, and Karl Johan Åström. "Measurement noise filtering for PID controllers." Journal of Process Control 24.4 (2014): 299-313.
> >
> > [15] Donoho, David L., Arian Maleki, and Andrea Montanari. "The noise-sensitivity phase transition in compressed sensing." IEEE Transactions on Information Theory 57.10 (2011): 6920-6941.
> >
> > [16] Wang, Bin, et al. "Recovery error analysis of noisy measurement in compressed sensing." Circuits, Systems, and Signal Processing 36 (2017): 137-155.
> >
> > [17] Iwen, Mark A., et al. "On recovery guarantees for one-bit compressed sensing on manifolds." Discrete & computational geometry 65 (2021): 953-998.
> >
> > [18] Xu, Weiyu, and Babak Hassibi. "Compressed sensing over the Grassmann manifold: A unified analytical framework." 2008 46th Annual Allerton Conference on Communication, Control, and Computing. IEEE, 2008.
> >
> > [19] Mai, Lihao, Haoran Li, and Yang Weng. "Data Imputation with Uncertainty Using Stochastic Physics-Informed Learning." 2024 IEEE Power & Energy Society General Meeting (PESGM). IEEE, 2024.

---

> ### Author Response · Authors · 2024-12-03
>
> $\textbf{Response:}$
>
> We greatly appreciate your thoughtful comments and your acknowledgment that the work is sound, along with the score increase. Below, we address your concerns regarding the study scope, representative experiments, and the expansion of Figure 4. While the paper revision deadline has passed, we will make every effort to incorporate these updates into the final version if the paper has the chance to be accepted.
>
> $\textbf{Narrow the Scope of the Paper}$.
>
>  In our last revision, we narrowed the scope by comprehensively modifying $\textbf{the Introduction}$, $\textbf{Assumption 1}$, $\textbf{Section 3.3 (Theoretical Analysis)}$, $\textbf{the Experiment}$, $\textbf{Conclusion and Future Work Section}$, and $\textbf{Appendix B}$.
>
> In this revision, we emphasize the scope in the statement of the proposed method. Hence, the study scope is $\textbf{complete and clear for the whole paper}$. Specifically,
>
> (1) After introduction Optimization (2), we add the following sentence. "Optimization (2) emphasizes the parameter alignment to represent HQ and LQ measurements with limited noise."
>
> (2) To describe Figure 1, I state that " Figure 1 illustrates how HQ and LQ parameters are aligned to capture the measurement correlations with limited random noise."
>
> (3) In Proposition 2, I mentioned that "Suppose Assumption (1) holds".
>
> (4) After Corollary 1 that states the global optimal solution for the alignment, we add "The alignment captures the HQ and LQ data correlations for weakly nonlinear systems and limited noise. In particular, in Proposition 3, we demonstrate the methods' robustness to linear Gaussian noise. The impact of other nonlinear noise needs further investigations."
>
> $\textbf{Representativeness of Table 1 and Table 2}$.
>
> Our experiments in Tables 1 and 2 are representative for several reasons.
>
> (1) $\textbf{Dataset Diversity}$. The datasets span a variety of systems influenced by human consumption behaviors (load data), weather patterns (PV/air quality data), and events (event synchrophasor data). Moreover, a continuous ODE system (Spiral data) is also considered.
>
> (2) $\textbf{Different data incompleteness scenarios are considered}$, including low resolutions, missing data, and irregularly sampled data.
>
> (3) $\textbf{Baseline models are comprehensive}$. In general, discrete sequence models (RNN), continuous models (ODE-RNN, Neural CDEm MFN), and parameter flow-based models (Neural ODE + RNN/Neural ODE + MFN) are comprehensively utilized.
>
> (4) $\textbf{Tasks are complete}$. We consider both interpolation and extrapolation tasks for time series.
>
> $\textbf{Expansion of Fig. 4}$.
>
> We follow your suggestion to conduct additional tests to expand experiments in Fig. 4. Specifically, we consider the additional LQ data coverage rate: $30\\%, 40\\%, 50\\%$. These tests are sufficient to demonstrate sensitivity with respect to the coverage rates. In general, $\textbf{the results are consistent with the results we already present in Figure 4}$. We present the MAPE (\%) for additional tests as follows.
>
> $\textbf{For interpolation}$:
>
> LQ data rate 30\%, 40\%, 50\%
>
> G-AlignNetR $\textbf{7.28\\%}$, $\textbf{4.55\\%}$, $\textbf{3.13\\%}$
>
> G-AlignNetI 8.62\%, 7.36\%, 5.65\%
>
> Linear Spline 10.29\%, 8.65\%, 6.91\%
>
> Cubic Spline 10.13\%, 8.91\%, 6.44\%
>
> CS 11.24\%, 9.38\%, 7.13\%
>
> DCS 18.45\%, 16.42\%, 14.78\%
>
> Semi-NN 12.45\%, 10.35\%, 8.12\%
>
> MFN 16.24\%, 13.13\%, 10.24\%
>
>
>
> $\textbf{For Extrapolation}$:
>
> LQ data rate 30\%, 40\%, 50\%
>
> G-AlignNetR $\textbf{8.95\\%}$, $\textbf{8.25\\%}$, $\textbf{7.69\\%}$
>
> G-AlignNetI 11.51\%, 11.12\%, 10.76\%
>
> MFN 14.92\%, 14.83\%, 13.59\%
>
> RNN 12.01\%, 11.76\%, 11.52\%
>
> ODE-RNN 12.16\%, 11.91\%, 11.69\%
>
> Neural CDE 12.05\%, 11.79\%, 11.53\%
>
> Neural ODE+RNN 12.59\%, 12.27\%, 11.96\%
>
> Neural ODE+MFN 13.27\%, 12.50\%, 12.03\%
>
> These results demonstrate the strong performance improvements for our proposed models due to our $\textbf{innovative geometric representation learning embedded with geometric optimization for assured quality alignment}$.
>
> We sincerely thank you for your feedback. All necessary updates will be incorporated into the camera-ready version if the paper is accepted, and we hope this revision fully addresses your concerns.

---

### Official Review · Reviewer_w1So · 2024-11-04

**Soundness:** 2
**Presentation:** 2
**Contribution:** 2
**Rating:** 5
**Confidence:** 1

**Summary:**

The paper proposes a G-align net framework that unifies high and low data modeling in parameter manifolds. The paper provides a theoretical guarantee for the performance and also empirically shows that it can outperform other baselines.

**Strengths:**

Originality: The idea of leveraging parameter geometry to enhance learning dynamics is novel.

Clarity: The paper includes several math propositions and some clear math derivations, though I cannot verify if they are correct.

Quality: The paper provides comprehensive results on several datasets for interpolation and extrapolation tasks. Most of the results can be output as baseline models.

Significance: It looks like the proposed framework could better preserve the geometry of the dataset and could also be used for some control tasks, in addition to better interpolation/extrapolation performance. However, I don't quite understand the significance of the work, and the author is welcome to illustrate it further in the case of the application value for physics systems.

**Weaknesses:**

The experiment is not always the best. And Figure 3 is confusing. Also training cost is not included for a better comparison. The dataset looks pretty simple so it is hard to evaluate its performance on complex physics dataset.

**Questions:**

1. In Figure 3, what does the green dot mean? The legend didn't mention the green dot where section 4.2 mentioned it. Moreover, the predictions don't align well with the truth in the rightmost column.

2. In Table 1, G-Align Net doesn't always perform the best. Could you further improve the result or explain it?

3. Could you include the training cost for all the algorithms?

4. Can a more complex dataset like weather/fluid dynamics be included?

5. A general setting about the LR and HR data. The paper mentioned that the amount of LR data is far less than that of HR data. However, in my understanding, the LR data is of low quality and is supposed to be easier to get than high-quality data. Could you elaborate on why your LR data is far less than HR data?

---

> ### Author Response · Authors · 2024-11-22
>
> $\textbf{W1. All questions}$.
>
> $\textbf{Response}$:
>
> Thank you for your insightful comment. We have provided a detailed response in the Questions section and have significantly improved the paper's clarity. We sincerely appreciate your efforts in highlighting these areas for improvement. Please refer to the revised version of the paper, and let us know if you have any further questions or suggestions.
>
> $\textbf{Q1. Fig. 3: legend and results}$.
>
> $\textbf{Response}$:
>
>  Thanks for your questions. We provide detailed answers and add key modifications to the revised paper.
>
> $\textbf{ What are the green dots?}$ In Fig. 3, the green dots in the left part of the Fig. represent the parameters $\Theta_y(t_i)$ corresponding to the low-resolution data points. The green dots in the right panel of Fig. 3 represent the low-resolution measurements that are used in the training procedure. High-resolution measurements are not visualized as they are too dense to display effectively. We add this legend to Fig. 3.
>
> $\textbf{Why misalignment in rightmost column?}$ Regarding the rightmost column, the discrepancy between the low-resolution prediction (dashed green) and the low-resolution ground truth (solid green) arises because the available data is limited to the discrete green dots, rather than the entire green curve. This sparsity of data inherently impacts the model's ability to perfectly align predictions with the true low-resolution values, and such alignment errors are a reasonable consequence of the insufficient data volume. However, it's obvious that with such limited low-resolution data, our G-AlignNet still performs better than the other method that doesn't perfectly align the structure of the parameter flow.
>
> The results demonstrate that G-AlignNet excels in aligning high-resolution and low-resolution measurements, extracting shared knowledge, and constructing a more accurate dynamic learning model for low-resolution data (as shown in the right panel of Fig. 3). This capability arises from G-AlignNet's ability to achieve precise parameter flow alignment (left panel of Fig. 3), which maximizes the extraction of common knowledge. The success of this alignment is attributed to our geometric representation learning approach, which leverages a well-structured parameter manifold, i.e., the orthogonal group.
>
> We hope this clarifies your concern and appreciate your attention to detail. We make all modifications accordingly in the revised paper.
>
> $\textbf{Q2. Explanations of Table 1}$.
>
> $\textbf{Response}$: Thank you for your observation. We acknowledge that in the interpolation task, there are certain scenarios where Deep Compressed Sensing (DCS) and cubic spline achieve slightly better performance. Below, we provide a detailed analysis. We also added the description below to the revised paper.
>
> $\textbf{Scenarios when cubic spline excels}$. Cubic spline demonstrates superior performance when the data drop rate is low, and the measurements exhibit smooth behavior. Under these conditions, the polynomial model inherent to cubic spline is well-suited to accurately fit the data. However, for complex systems with higher uncertainty or when the data drop rate is significant (e.g., in low-resolution data scenarios), cubic spline fails to generalize effectively, limiting its utility.
>
> $\textbf{Scenarios when DCS excels}$. DCS [1] employs a pre-trained variational autoencoder to approximate the distribution of high-quality (HQ) data, using it to perform interpolation. This approach works particularly well when HQ and low-quality (LQ) data exhibit minimal distribution shift. For instance, in photovoltaic (PV) and air quality systems, both HQ and LQ data are typically collected within the same local region under consistent weather conditions, resulting in similar measurements for solar generation and air quality. As a result, DCS achieves high accuracy. However, for datasets where HQ and LQ data exhibit significant differences, DCS struggles to maintain performance.
>
> In contrast, G-AlignNet effectively extracts shared underlying structures across datasets by modeling them as parameter flow shapes. The embedded geometric optimization ensures that even when there is high uncertainty or significant disparity between HQ and LQ data, all available information is utilized efficiently to achieve superior interpolation of LQ data. This robustness highlights the strength of G-AlignNet in handling challenging scenarios where other methods falter.

---

> ### Author Response · Authors · 2024-11-22
>
> $\textbf{Q3. Training cost}$.
>
> $\textbf{Response}$:
>
> Thanks for your question. We have included the training time (minutes) in Table 3 in Appendix D.3. The results show that our methods, with a Runge-Kutta 4 (RK4) ODE solver and a relatively high tolerance, can achieve relatively moderate training time and the best model performance.
>
> $\textbf{Q4. Complex datasets}$.
>
> $\textbf{Response}$:
>
> Thank you for your insightful comments.
>
> $\textbf{1. Our test datasets are already complex.}$ In this study, we focus on engineering and control systems characterized by high uncertainty. For example:
>
> 1.1 Residential load data: This represents household electricity consumption, which is influenced by both weather conditions and human behaviors.
>
> 1.2 Photovoltaic (PV) generation: PV output is governed by solar and wind patterns, adding dynamic complexity.
>
> 1.3 Power events: These involve transient processes (lasting less than 10 seconds) with high-frequency oscillations.
>
> These systems are inherently challenging and have been studied for decades to develop effective forecasting models [2,3]. As demonstrated in Table 2, our G-AlignNet outperforms numerous advanced baselines for short-term forecasting, especially in scenarios with low-quality data.
>
> $\textbf{2. Including PDE-based systems is a promising future direction}$. We agree that partial differential equation (PDE) systems with complex spatiotemporal correlations present an important area for further exploration. Such systems also face significant challenges related to data quality [4,5]. We believe that G-AlignNet could be effectively applied to capture shared spatiotemporal structures within PDE measurements.
>
> Furthermore, as noted in Section 3.2, the geometric representations learned by G-AlignNet could be utilized to shape parameters for Implicit Neural Representations (INRs) and Physics-Informed Neural Networks (PINNs). While this paper focuses on engineering systems and provides sufficient technical contributions, numerical tests, and theoretical analyses, we consider extending G-AlignNet to PDE systems as a promising avenue for future work.
>
> $\textbf{Q5. HR/LR data setting}$.
>
> $\textbf{Response}$:
>
> Thank you for your insightful question. We appreciate the opportunity to clarify this point. We add the below description to Section 3.1 of the paper.
>
> We agree that LR data are generally easier to obtain, as systems often contain more LR sensors than HR sensors. This means that the dimensionality of LR measurements ($\boldsymbol{y}\in\mathbb{R}^{d_y}$) is typically greater than the dimensionality of HR measurements ($\boldsymbol{x}\in\mathbb{R}^{d_x}$), i.e., $d_x<d_y$.
>
> However, when considering the number of samples collected within a fixed time interval, HR data often outweigh LR data due to differences in sampling frequency. HR sensors have significantly smaller sampling intervals, allowing them to record more measurements over time. As a result, the number of HR samples ($|\mathcal{N}_x|$) is greater than the number of LR samples ($|\mathcal{N}_y|$), i.e., $|\mathcal{N}_x|>|\mathcal{N}_y|$. This distinction between data dimensionality and sampling frequency explains why HR data can be more abundant than LR data, even though LR sensors are more prevalent and easier to deploy. We give clear statements in Section 3.1.
>
> $\textbf{References}$:
>
> [1] Y. Wu, M. Rosca, and T. Lillicrap, “Deep compressed sensing,” in International Conference on Machine Learning. PMLR, 2019, pp. 6850–6860.
>
> [2] H. K. Alfares and M. Nazeeruddin, “Electric load forecasting: literature survey and classification of methods,” International Journal of Systems Science, vol. 33, no. 1, pp. 23–34, 2002.
>
> [3] R. Ahmed, V. Sreeram, Y. Mishra, and M. Arif, “A review and evaluation of the state-of-the-art in pv solar power forecasting: Techniques and optimization,” Renewable and Sustainable Energy Reviews, vol. 124, p.109792, 2020.
>
> [4] Maddu, Suryanarayana, et al. "Stability selection enables robust learning of partial differential equations from limited noisy data." arXiv preprint arXiv:1907.07810 (2019).
>
> [5] Zhang, Zhiming, and Yongming Liu. "A robust framework for identification of PDEs from noisy data." Journal of Computational Physics 446 (2021): 110657.

---

> > ### Author Response · Authors · 2024-11-30
> > **Follow-Up on Rebuttal and Clarifications Regarding Reviewer Concerns**
> >
> > Dear Reviewer w1So:
> >
> > I hope you are doing well. I would like to thank you once again for your feedback on our paper and for raising thoughtful questions regarding the problem setting, result explanation, and clarity. Your input has been valuable in helping us identify areas where further clarification was needed. We have provided a detailed response in our rebuttal, aiming to address your concerns comprehensively and enhance the overall presentation of our work. I understand that this domain might be outside your primary area of expertise, and I greatly appreciate the time and effort you’ve taken to engage with it. It’s worth noting that the other reviewers have expressed positive opinions on the work, which might provide additional context as you revisit the discussion.
> >
> > I recognize that this is a busy time of year, and I hope you were able to enjoy the recent Thanksgiving holiday. If you could take a moment to review our response and share your thoughts when convenient, we would deeply value your feedback. Please let us know if further clarification or supporting information would help facilitate your evaluation.

---

### Author Response · Authors · 2024-11-22
**Response Summary and Revisions Overview for All Reviewers and Editors**

We would like to thank all reviewers and editors for their thoughtful feedback and insightful questions, which have helped us greatly improve the quality of our manuscript. With your help, we have successfully $\textbf{submitted a revised manuscript}$, and all changes are highlighted in blue for easy reference. In particular, we make significant improvements in the following aspects:

1. Clearly defining data quality issues, assumptions, and problem formulation, ensuring consistency throughout the paper.

2. Discussing the limitations of existing methods in handling data quality issues and dynamic model learning, with numerical and theoretical support, emphasizing the motivation behind our G-AlignNet's design.

3. Improving the clarity of theoretical proofs, particularly Proposition 4, by explicitly addressing discretization errors.

4. Providing additional experimental results, including training time comparisons, evaluations for G-AlignNetI (using INR as the base model), verification of the orthogonality of parameter flows, etc.

5. Expanding on experimental details and clarifying all unclear experimental presentations (e.g., green points in Figure 3), as well as providing dataset dimensions and settings.

6. Exploring the feasibility of extending G-AlignNet to other sequence models, such as Transformers and Mamba, while acknowledging computational trade-offs.

We sincerely hope the revisions address the reviewers’ concerns comprehensively and are welcome to further discussions to clarify or refine any points further. Thank you again for your time and efforts in reviewing our work!

---

### Note · Authors · 2025-01-23

I have read and agree with the venue's withdrawal policy on behalf of myself and my co-authors.